# A Unified Approach Towards Active Learning and Out-of-Distribution Detection

**Sebastian Schmidt***                                            *sebastian95.schmidt@tum.de*
*Technical University of Munich*
*BMW Group*

**Leonard Schenk***                                               *leonard.schenk@sprind.org*
*Sprin-D*

**Leo Schwinn**                                                   *l.schwinn@tum.de*
*Technical University of Munich*

**Stephan Günnemann**                                            *s.guennemann@tum.de*
*Technical University of Munich*

**Reviewed on OpenReview:** *https://openreview.net/forum?id=HL75La10FN*

## Abstract

In real-world applications of deep learning models, active learning (AL) strategies are essential for identifying label candidates from vast amounts of unlabeled data. In this context, robust out-of-distribution (OOD) detection mechanisms are crucial for handling data outside the target distribution during the application's operation. Usually, these problems have been addressed separately. In this work, we introduce SISOM as a unified solution designed explicitly for AL and OOD detection. By combining feature space-based and uncertainty-based metrics, SISOM leverages the strengths of the currently independent tasks to solve both effectively without requiring specific training schemes. We conducted extensive experiments showing the problems arising when migrating between both tasks. In our experiments SISOM underlined its effectiveness by achieving first place in one of the commonly used OpenOOD benchmark settings and top-3 places in the remaining two for near-OOD data. In AL, SISOM[1] delivers top performance in common image benchmarks.

## 1 Introduction

Large-scale deep learning models encounter several data-centric challenges during training and operation, particularly in real-world problems such as mobile robotic perception (Cai & Koutsoukos, 2020) and autonomous driving (Nitsch et al., 2021). On the one hand, these models require vast amounts of data and labels for training, driven by the uncontrolled nature of real-world tasks. On the other hand, even when trained with extensive data, these models can behave unpredictably when facing samples that deviate significantly from training data (Schmidt et al., 2025a), known as out-of-distribution (OOD) data.

Active learning (AL) addresses the first limitation by guiding the selection of label candidates. In the traditional pool-based

Figure 1: CIFAR-10 UMAP plot of unlabeled, near, and far OOD data compared to labeled data. For details, see Appendix B.4.

---

*Equal Contribution
[1]Project Page with Code: https://www.cs.cit.tum.de/daml/sisom

AL scenario (Settles, 2010), models start with a small labeled training set and can iteratively query data and its labels from an unlabeled data pool. The selection is based on model metrics such as uncertainty, diversity, or latent space encoding. One cycle concludes with the training on the increased labeled subset.

The second challenge, dealing with unknown data during operation, is typically addressed by OOD detection. OOD detection distinguishes between in-distribution (InD) data used for training the model and OOD samples, which differ from the training distribution. Literature differentiates between near-ODD and far-OOD, which can be categorized by the type of distribution shifts occurring. Yang et al. (2022); Zhang et al. (2023) define near-OOD as a pure covariate shift, while far-OOD often contains a semantic shift.

Recent works in AL combined AL and OOD detection into the Open-set AL (OSAL) scenario, assuming the existence of OOD data in the unlabeled pool, enabling a selection from the data pool without prior knowledge. However, existing methods (Ning et al., 2022; Yang et al., 2023; Du et al., 2021) usually rely on separate components for OOD separation and data selection, respectively.

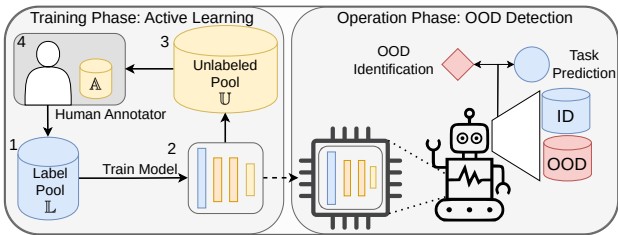

Figure 2: Real-world application life cycle comprising active learning in the training phase (left) and out-of-distribution detection in the operation phase (right).

As a result, the connection between both tasks has rarely been explored. Methodologically, both utilize common metrics, such as uncertainty, latent space distances, and energy. In addition, a sample detected by such metrics can be, on the one hand, a novel AL sample that is insufficiently represented by the current training distribution. On the other hand, the sample can pose a covariant shift in an OOD setting. Considering both cases, as depicted in Fig. 1, shows an ambiguity and overlap of both sample categories. This raises the question whether an examination of the ambiguity of and relations between the respective samples can provide valuable insights for designing approaches for both tasks and OSAL , where current methods mostly consider both subtasks as independent.

Besides the methodology perspective, mobile robotic applications often train neural networks on recordings and afterward deploy them for operation as shown in Fig. 2 as an integrated life cycle. Given the amount of collected data, AL is applied for label-efficient training. During operation, OOD detection is employed to observe if the current operation is within the trained domain, which is necessary for real-world operation domains. Existing works address both challenges with separate methods that can lead to diverging goals like specific training schemes. Addressing these tasks separately introduces significant overhead, especially for deployment and development, such as hyperparameter optimization or the training of auxiliary models.

*Our work* explores the connection between AL and OOD detection, introducing a unified approach for both tasks that leverages their mutual strengths. Specifically, we propose **S**imultaneous **I**nformative **S**ampling and **O**utlier **M**ining (**SISOM**), which uses enriched feature space distances based on coverage creating a symbiosis between AL and OOD detection. By exploiting the ambiguity of both tasks, SISOM achieves top performance across most near-OOD and AL benchmarks. By uniting both tasks, SISOM simplifies the application life cycle by *eliminating the need for a separate OOD design phase* and resolves conflicting design goals. This perspective contrasts with open-set AL, where OOD data is incorporated into the unlabeled pool, forming a combined task. Additionally, SISOM provides a *novel latent space analysis* for *post-training latent space refinement* and a first-of-its-kind *self-balancing of uncertainty and diversity metrics*. In summary, *our contributions* are as follows:

- We propose **S**imultaneous **I**nformative **S**ampling and **O**utlier **M**ining (**SISOM**), a novel method designed for both OOD detection *and* AL.

- We introduce a latent space analysis enabling an *optimization loop* for further *post-training latent space refinement* and a *self-balanced uncertainty diversity fusion*.

- In extensive experiments, we demonstrate SISOM effectiveness in common image AL *and* OOD benchmarks against highly specialized state-of-the-art methods.

## 2 Preliminaries

**Active Learning:** AL is a subfield of machine learning designed to reduce the number of required labels by querying a set of new samples $\mathbb{A}$ of a query size $q$ in a cyclic process. Let $\mathcal{X}$ represent a set of samples and $\mathcal{Y}$ a set of labels. AL starts with an initially labeled pool $\mathbb{L}$, containing data samples with features $\mathbf{x}$ and corresponding label $y$, and an unlabeled pool $\mathbb{U}$ where only $\mathbf{x}$ is known. However, $y$ can be queried from a human oracle. We further assume that $\mathbb{L}$ and $\mathbb{U}$ are samples from a distribution $\Omega$. In each cycle, a model $f$ is trained such that $f : \mathcal{X}_{\mathbb{L}} \to \mathcal{Y}_{\mathbb{L}}$. This model then selects new samples from $\mathbb{U}$ based on a query strategy $Q(\mathbf{x}, f)$, which utilizes (intermediate) model outputs. As a result, the newly annotated set $\mathbb{A}$ is added to the labeled pool $\mathbb{L}^{i+1}$ and removed from the unlabeled pool $\mathbb{U}^{i+1}$.

**Out of Distribution Detection:** Ancillary, OOD detection assumes a model $f : \mathcal{X}_{\mathbb{L}} \to \mathcal{Y}_{\mathbb{L}}$ trained on our training data $\{\mathbf{x}, y\} \in \mathbb{L}$ which have been sampled from the distribution $\Omega$. During evaluation or inference, a model $f$ encounters data samples $\tilde{\mathbf{x}}$ from a distribution $\Theta$ and $\Omega$, where $\Omega \cap \Theta = \emptyset$ and $\tilde{\mathbf{x}} \notin \mathbb{L}$. Data sampled from $\Omega$ are referred to as InD data, while samples from $\Theta$ are referred to as OOD data. Based on the trained model $f$, a metric $S$ is used to determine whether a sample $x$ is sampled from $\Omega$ or $\Theta$.

$$G(\mathbf{x}, f) = \begin{cases} \text{InD} & \text{if } S(\mathbf{x}; f) \geq \lambda \\ \text{OOD} & \text{if } S(\mathbf{x}; f) < \lambda \end{cases} \tag{1}$$

OOD detection is further categorized into near- and far-OOD (Zhang et al., 2023). Far-OOD refers to completely unrelated data, such as comparing MNIST (LeCun et al., 1998) to CIFAR-100 (Krizhevsky et al., 2009), while CIFAR-10 (Krizhevsky et al., 2009) to CIFAR-100 would be considered as near-OOD. OpenODD (Yang et al., 2022) ranks near-OOD detection as more challenging.

## 3 Related Work

**Active Learning:** AL mainly considers the pool-based and stream-based scenario (Settles, 2010), where data is either queried from a pool in a data center or a stream on the fly. For deep learning, the majority of current research deals with pool-based AL (Ren et al., 2021). However, further scenarios have been evaluated by Schmidt & Günnemann (2023) and Schmidt et al. (2024). Independent of the scenarios, samples are selected either by prediction uncertainty, latent space diversity, or auxiliary models. A majority of the uncertainty-based methods rely on sampling - like Monte Carlo Dropout (Gal & Ghahramani, 2016) - or employ ensembles (Beluch et al., 2018; Lakshminarayanan et al., 2017). To additionally ensure batch diversity Kirsch et al. (2019) used the joint mutual information. The uncertainty concepts have been employed and further developed for major computer vision tasks, including object detection (Feng et al., 2019; Schmidt et al., 2020), 3D object detection (Hekimoglu et al., 2022; Park et al., 2023b), and semantic segmentation (Huang et al., 2018). One of the few works breaking the gap between both tasks (Shukla et al., 2022) modified an OOD detection method for pose estimation. Dutta et al. (2025), proposed to use imprecise probability by Imprecise Neural Networks for statistical verification of autonomous systems to enable safe operation and AL for reinforcement learning. Mukhoti et al. (2023) proposed DDU, an uncertainty baseline based on spectral convolutions and Gaussian mixture models, which show improvements against other general uncertainty approaches on AL and OOD detection compared to other uncertainty approaches. Given the requirements of spectral convolutions, DDU is not flexible applicable for all use cases. In contrast, diversity-based approaches aim to select key samples to cover the whole dataset. Sener & Savarese (2018) proposed to choose a CoreSet of the latent space using a greedy optimization. Yehuda et al. (2022) selected samples having high coverage in a fixed radius for low data regimes. Mishal & Weinshall (2024) extended the approach for more data regimes dynamic strategy mixing. Ash et al. (2020) enriched the latent space dimensions to the dimensions of the gradients and included uncertainty in this way. The concept of combining uncertainty with diversity has been further refined for 3D object detection (Yang et al., 2024; Luo et al., 2023). Liang et al. (2024) combined different diversity metrics for the same task. In semantic segmentation, Surprise Adequacy (Kim et al., 2020) has been employed to measure how surprising a model finds a new instance. Yi et al. (2022) used auxiliary tasks for unsupervised model training to select diverse samples. Besides the metric-based approach, the selection can also be made by auxiliary models mimicking diversity and uncertainty. These

approaches range from loss estimation (Yoo & Kweon, 2019), autoencoder-based approaches (Sinha et al., 2019; Zhang et al., 2020; Kim et al., 2021) and graph models (Caramalau et al., 2021), to teacher-student approaches (Peng et al., 2021; Hekimoglu et al., 2024).

**Out-of-Distribution Detection:** To facilitate a fair comparison and evaluation of OOD methods, benchmarking frameworks like OpenOOD (Yang et al., 2022; Zhang et al., 2023) have been introduced, which categorizes the methods into preprocessing methods altering the training process and postprocessing methods being applied after training. Preprocessing techniques include augmenting training data like mixing (Zhang et al., 2018; Tokozume et al., 2018) different samples or applying fractals to images (Hendrycks et al., 2022). Postprocessing approaches include techniques of manipulations on neurons and weights of the trained network, such as filtering for important neurons (Ahn et al., 2023; Djurisic et al., 2022), or weights (Sun & Li, 2022), or clipping neuron values to reduce OOD-induced noise (Sun et al., 2021). Recent advancements in post-hoc network enhancement include SCALE (Xu et al., 2024), which enhances OOD detection by activation scaling. Logit-based approaches encompass the model output to estimate uncertainties using temperature-scaling (Liang et al., 2018), modified entropy scores (Liu et al., 2023), energy scores (Liu et al., 2020; Elflein et al., 2021) or ensembles (Arpit et al., 2022). Other methods use distances in the feature space, such as the Mahalanobis distance between InD and OOD samples (Lee et al., 2018), rely on gradients (Liang et al., 2018; Hsu et al., 2020; Huang et al., 2021; Schwinn et al., 2021), estimate densities (Charpentier et al., 2020; 2022) or apply k-nearest-neighbor on latent space distances (Sun et al., 2022). Nearest Neighbor Guidance (NNGuide) (Park et al., 2023a) refines classifier-based scores by respecting the data manifold's boundary geometry. A different branch operates on the features directly and evaluates properties like the Norm (Yu et al., 2023) or performs rank reductions via SVD (Song et al., 2022). NAC (Liu et al., 2024) combined gradient information with a density approach, where a probability density function over InD samples is estimated. CombOOD (Rajasekaran et al., 2024) is a semi-parametric framework that combines nearest-neighbor and Mahalanobis distances to improve OOD detection accuracy.

**OpenSet Active Learning:** The emerging field of OpenSet AL considers both tasks in one cycle, assuming the AL pool is polluted by OOD samples. Existing approaches (Ning et al., 2022; Park et al., 2022; Yang et al., 2023; Safaei et al., 2024) present strong results by primarily tackling both tasks by *separate* modules containing auxiliary models. As unlabeled and OOD samples are mostly considered decoupled with uncorrelated modules, this field is orthogonal to our examination of correlation and entanglement. Alternative approaches like SIMILAR (Kothawade et al., 2021) utilize submodular information measures to handle OOD data in unlabeled sets, adding computational overhead and needing initial access to OOD data. Additionally, Stojnić et al. (2024) expanded the open-set classifier to an ensemble of models with an extra OOD class for semi-supervised AL, increasing computational demands. In contrast to works in this field, we investigate the correlation between unlabeled and OOD samples to provide a unified metric for both. We believe that this field benefits from the joint consideration of AL and OOD samples and an examination of their ambiguity.

While various works exist in OOD and AL, both tasks are considered independent. Even in OSAL, the tasks are **mostly** considered by independent method components. Some uncertainty methods are evaluated on both tasks but limit their evaluation to the uncertainty domain. Current state-of-the-art approaches are often specified for one task. In addition, the application life cycle consideration is unexplored.

## 4   Methodology

To address both AL and OOD detection tasks in a unified method to simplify real-world applications, we need to first understand the goals of these two tasks. AL aims to identify and select samples that are beneficial for training and increase the model's performance. These samples typically position themselves between the existing clusters in the latent space or near the decision boundaries. OOD detection targets the identification of data outside the training data and, therefore, outside the known clusters in latent space. Given the definition of far- and near-OOD, near-ODD is closer to InD data and located close to the decision boundaries and in between the existing clusters. Liu & Qin (2024) recently showed that OOD is generally closer to the decision boundary than InD, confirming this hypothesis. Fig. 1 investigates this hypothesis and shows the overlap in distribution of interesting unlabeled data and (near-)OOD data compared to the distance to far-OOD samples.

To target these overlapping regions, we designed a method that focused on the latent space regions between the clusters. To do so, SISOM employs an enlarged feature space **Coverage (1)** and increases expressiveness by weighting important neurons in a **Feature Enhancement (2)**. Based on this feature representation, we refine the AL selection and the InD and OOD border by using an inner-to-outer class **Distance Ratio (3)**, guiding it to unexplored and decision boundary regions. As feature space distances are prone to poorly defined latent space representations, we introduce **Feature Space Analysis (4)**, providing a self-deciding fusion of our distance metric with an uncertainty-based energy score. Optionally, our previous analysis enables us to optimize the **Sigmoid Steepness (5)**, providing a further refinement of the feature space representations from **(2)**. An overview is depicted in Fig. 3. With this setup, SISOM specifically addresses both tasks without any requirements on the training scheme.

**(1) Coverage:** We aim to identify the regions of the samples that are interesting and unexplored for AL, as well as OOD samples in latent space. To do so, we rely on an informative latent space covering as much information as possible.

To increase the information gain, we cover the full network and define the feature space representation of an input sample $\mathbf{x}$ as a concatenation of the latent space of multiple layers $h_j$ in a set of layers $H$ in Eq. (2). This follows the procedures of neural coverage (Kim et al., 2019; Liu et al., 2024) and is contrasting to most diversity-based AL approaches (Sener & Savarese, 2018; Ash et al., 2020), which use a single layer.

$$\mathbf{z} = h_1(\mathbf{x}) \oplus \cdots \oplus h_j(\mathbf{x}) \oplus \cdots \oplus h_n(\mathbf{x}) \tag{2}$$

Given the feature space $\mathbf{z}$, we further denote $\mathbb{Z}_U$ as a set of feature space representations of unlabeled samples from $\mathbb{U}$, while $\mathbb{Z}_L$ denotes the set of representations of all labeled samples $\mathbb{L}$.

**(2) Feature Enhancement:** To enhance the expressiveness of our defined latent space, improving class separation, we introduce a weighting of individual layers. Prior research (Huang et al., 2021; Liu et al., 2024) has demonstrated that the gradients of neurons with respect to the KL divergence of the model's output and a uniform distribution encapsulate valuable information for OOD detection.

We apply the technique to improve the features further and enrich these by representing the individual contribution of each neuron $i$, denoted as $g_i$. This gradient describes each neuron's contribution to the actual output being different from the uniform distribution. A low value suggests that the neuron has little influence on the prediction of a given input sample. Conversely, if the value is high, the respective neuron is crucial for the decision process.

Thus, the gradient vector can be interpreted as a saliency weighting for the activation values in the feature space to support separability. In detail, we compute the gradient of the Kullback-Leibler (KL) divergence between an uniform distribution $u$ and the softmax output distribution $f(\mathbf{x})$ for an input sample $\mathbf{x}$ with respect to the selected features $\mathbf{z}$:

$$\mathbf{g} = \frac{\partial D_{KL}(u||f(\mathbf{x}))}{\partial \mathbf{z}}. \tag{3}$$

We create a weighted feature representation $\tilde{\mathbf{z}}$ by multiplying the calculated saliency element-wise with the feature space representation $\mathbf{z}$ using the hadamard product $\odot$ and restrict it to $[-1, 1]$ with the sigmoid function $\sigma$:

$$\tilde{\mathbf{z}} = \sigma(\mathbf{z} \odot \mathbf{g}). \tag{4}$$

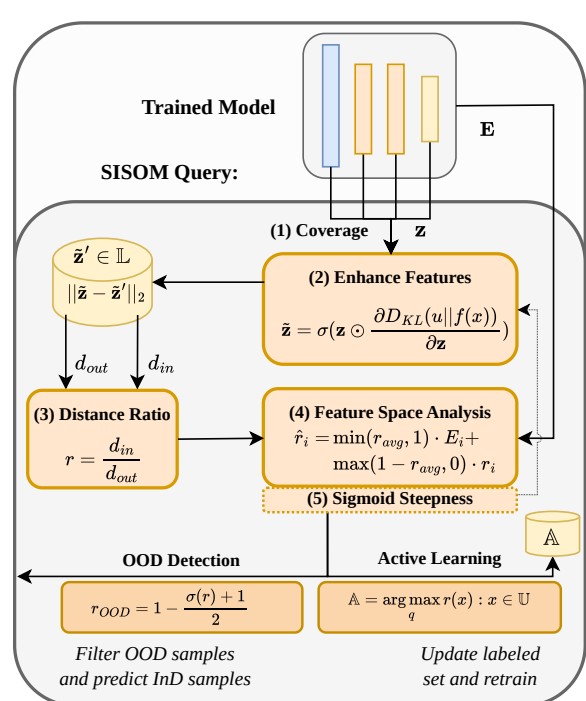

Figure 3: SISOM framework for OOD detection and AL combined.

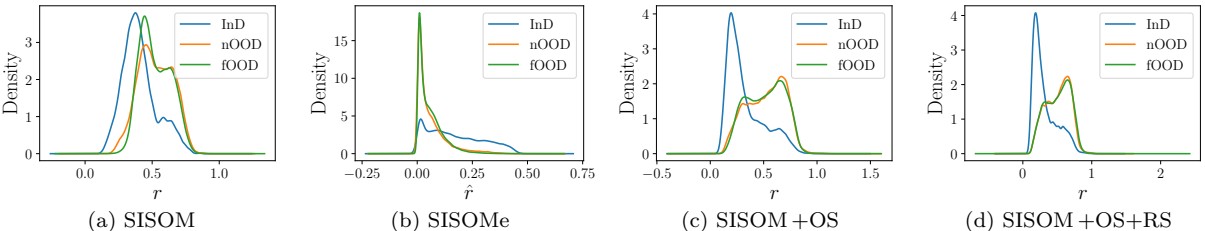

Figure 4: Density plots for SISOM with energy, Optimal Sigmoid Steepness (OS) and Reduced Subset Selection (RS) on CIFAR-100 with near-OOD (nOOD) and far-OOD (fOOD) as defined in OpenOOD.

The resulting gradient-weighted feature representation effectively prioritizes the most influential neurons for each input. This facilitates the identification of inputs activating atypical influence patterns, which is significant for AL as well as OOD detection. A qualitative analysis demonstrating the effect of the feature enrichment is given in Appendix A.4.

**(3) Distance Ratio:** After we defined and enhanced our latent space, we design our metric to identify the respective samples. Contrasting to other works in the latent space domain for AL and OOD detection (Sener & Savarese, 2018), which rely on simple distance metrics, we take inspiration from complex distance metrics (Kim et al., 2019) for detecting adversarial examples.

We assume the location of important samples is between the existing clusters in latent space. While far-OOD and curious AL samples are easier to detect due to larger latent space distances, near-OOD or AL samples close to the decision boundary are more difficult to detect and often more important. To identify samples in these regions, we rely on a distance quotient between inner-class and outer-class distances, boosting the detection of samples close to the decision boundary.

The inner-class distance $d_{in}$ is defined as the minimal feature space distance to a known sample of the same class $c$ as the predicted pseudo-class of the given sample. The outer-class distance $d_{out}$ represents the minimal feature space distance to a known sample of a different class than the sample's pseudo-class.

$$d_{in}(\tilde{\mathbf{z}}) = \min_{\mathbf{z}' \in \mathbb{Z}_L (c' = c)} ||\tilde{\mathbf{z}} - \tilde{\mathbf{z}}'||_2 \qquad (5) \qquad\qquad d_{out}(\tilde{\mathbf{z}}) = \min_{\mathbf{z}' \in \mathbb{Z}_L (c' \neq c)} ||\tilde{\mathbf{z}} - \tilde{\mathbf{z}}'||_2 \qquad (6)$$

The distance is computed on the gradient-enhanced feature space $\tilde{\mathbf{z}}$, defined in Eq. (4), with $\mathbf{z}'$ describing the nearest sample from the set of known samples $\mathbb{Z}_L$.

In many state-of-the-art works on AL, computationally expensive distance calculations are often present (Sener & Savarese, 2018; Ash et al., 2020; Caramalau et al., 2021). To make our approach more efficient for AL and feasible for large-scale OOD detection tasks, we select a representative subset $\mathbb{T} \subset \mathbb{Z}_L$ as a comparison set, thereby significantly reducing computational overhead. We utilize a fixed radius neighborhood to select samples, maximizing the coverage of the dataset within this radius in the feature space for each class. The effect of this subset selection is further investigated in Section 5.

Our SISOM score $r$ reflects the distance between each neuron's weighted feature representation in the latent space and the nearest sample of the predicted class relative to the closest distance to a sample from a different class for a sample $\mathbf{x}$ and the respective representation $\tilde{\mathbf{z}}$:

$$r(\mathbf{x}) = \frac{d_{in}}{d_{out}}. \qquad (7)$$

An extended comparison of the different distance metrics and their ability to separate InD and OOD is shown in Appendix A.4, while a SISOM is depicted in Fig. 4a. We further omit $\mathbf{x}$ for $r$ after Eq. (8).

Since we calculated a scale-value for each sample, we follow the most commonly used top-k selection (Yoo & Kweon, 2019; Kim et al., 2021; Gal & Ghahramani, 2016) to select $q$ samples with the highest distance ratio $r$, with $q$ being the AL query size:

$$\mathbb{A} = \operatorname{argmax}_q r(\mathbf{x}) : \mathbf{x} \in \mathbb{U}. \qquad (8)$$

For OOD Detection, we map the distance ratios $r$ to an interval $[0; 1]$ with the strictly monotonically decreasing function:

$$r_{OOD} = 1 - \frac{\sigma(r) + 1}{2}. \tag{9}$$

**(4) Feature Space Analysis:** Having a well-defined latent space is crucial for SISOM to attain optimal performance. Furthermore, we hypothesize that techniques relying on feature space metrics are more dependent on feature space separation than uncertainty-based methods. This dependency is important for SISOM as it utilizes a quotient of feature space metrics. Nevertheless, obtaining a well-defined and separable latent space may pose challenges in specific contexts and tasks. To estimate the separability of feature space, we compute the average distance ratio $r_{avg}$ using Eq. (4) and Eq. (7) for the known set as:

$$r_{avg} = \frac{1}{|\mathbb{L}|} \sum_{\tilde{\mathbf{z}} \in \mathbb{L}} \frac{d_{in}(\tilde{\mathbf{z}})}{d_{out}(\tilde{\mathbf{z}})} = \frac{1}{|\mathbb{L}|} \sum_{\mathbf{z} \in \mathbb{L}} \frac{d_{in}(\sigma(\mathbf{z} \odot \mathbf{g}))}{d_{out}(\sigma(\mathbf{z} \odot \mathbf{g}))}. \tag{10}$$

A lower $r_{avg}$ value indicates better separation of the samples in the enhanced feature space, implying that samples of the same class are relatively closer together than samples of different classes.

To mitigate possible performance disparities of SISOM in difficult separable domains, we introduce a novel self-deciding process combining SISOM with the uncertainty-based energy score $E(\mathbf{x}) = -\log \sum_{i=1}^{c} \exp(f(\mathbf{x})_i)$ estimated from the model's output logits $f(\mathbf{x})$. We utilize $r_{avg}$ to weights the combination of uncertainty with our SISOM diversity metric as follows for an instance $i$:

$$\hat{r}_i = \min(r_{avg}, 1) \cdot E_i + \max(1 - r_{avg}, 0) \cdot r_i. \tag{11}$$

The created SISOMe score $\hat{r}$ balances between uncertainty and diversity and relies either more on the energy score $E$ or the distance ratio $r_i$, based on the feature space separability. If $r_{avg} \to 1$, indicating poorly separated classes, $\hat{r}_i$ relies more on the energy score. Conversely, if $r_{avg} \to 0$, suggesting a well-separated feature space, $\hat{r}_i$ relies more on the distance ratio. A density outline of our combined approach SISOMe is given in Fig. 4b. Alternatively, one can replace $r_{avg}$ with a tuneable hyperparameter in Eq. (11).

**(5) Sigmoid Steepness:** Since Eq. (10) depends on the sigmoid function defined in Eq. (4), the sigmoid function has a large influence on the enhanced feature space $\tilde{\mathbf{z}}$. An additional hyperparameter $\alpha$ can influence the sigmoid function's steepness. As $\mathbf{z}$ is concatenated from different layers in Eq. (2), the sigmoid can be applied to each layer $j$ individually. This allows for a more nuanced control over the influence of each neuron's contribution to the final decision, and so influences the separability of the feature space. We define the sigmoid using the steepness parameter $\alpha$ as:

$$\sigma_j(\mathbf{x}) = \frac{1}{1 + e^{-\alpha_j \mathbf{x}}}; \quad \{\alpha_j : h_j \in \mathbf{z} \ \forall j\}. \tag{12}$$

Relating to Eq. (4), the set $\alpha$ of steepness parameters of the sigmoid function for each layer $h_j$, determines the degree of continuity or discreteness of the features within that layer. By applying a layerwise sigmoid, Eq. (4) is formulated as follows:

$$\tilde{\mathbf{z}} = \sigma_1(h_1(\mathbf{x}) \odot g_{i,1}) \oplus \cdots \oplus \sigma_j(h_j(\mathbf{x}) \odot \mathbf{g}_{i,j}) \oplus \cdots \oplus \sigma_n(h_n(\mathbf{x}) \odot \mathbf{g}_{i,n}),$$
$$\text{with} \quad \mathbf{g}_{i,j} = \frac{\partial D_{KL}(u||f(\mathbf{x}))}{\partial h_{j,i}}; \quad \forall j.$$

Following this consideration we can select $\alpha$ values which optimize the feature space separability metric $r_{avg}$ from Eq. (10) by minimizing $\alpha_{opt} = \arg\min_\alpha r_{avg}(\alpha)$. Besides the quantitative assessment of our Feature Space Analysis and Sigmoid Steepness in Section 5, the influence of Sigmoid Steepness is shown in Fig. 4c.

## 5 Experiments

To evaluate the abilities of SISOM for the real-world application life cycle (Fig. 2), we conducted comprehensive experiments on **AL** and **OOD detection** individually. We utilize the commonly used closed-set

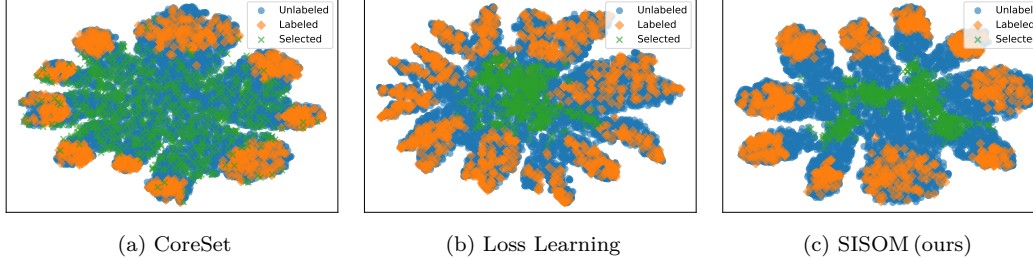

(a) CoreSet        (b) Loss Learning        (c) SISOM (ours)

Figure 5: T-SNE feature space comparison of Loss Learning, CoreSet, and SISOM for SVHN on cycle 1. SISOM effectively targets the areas in-between the clusters.

pool-based AL scenario (Settles, 2010) for AL. For OOD detection, we employ the extensive OpenOOD benchmark (Yang et al., 2022; Zhang et al., 2023). In addition, we conduct OOD detection using models trained by AL to evaluate the whole application life cycle steps from Fig. 2.

An evaluation of the **OSAL** scenario, where OOD data is present in the unlabeled AL pool, is out of scope in this work. Existing OSAL (Ning et al., 2022; Park et al., 2022; Yang et al., 2023; Safaei et al., 2024) approaches employ **two** components, one for each task, while this work focuses on a single component that solves both based on an ambiguity. This ambiguity would confuse the proposed method when applied independently to the compound task. In addition, the OOD filtering components often require access to OOD samples, which is not permitted in the classic OOD detection setup. However, a recent work (Schmidt et al., 2025b) effectively integrated SISOM with additional components into a framework addressing OSAL.

**Latent Space Assessment:** In Fig. 1, we compare near- and far-OOD data as well as unlabeled data with the labeled data. It can be seen that the near-OOD data and the unlabeled data are positioned close to the individual clusters of the InD labeled data. This confirms our hypothesis that near-OOD can be ambiguous to unlabeled data. Moreover, in real-world applications, the model is more likely to encounter semantic shifts along with contextually similar data, like Tiny ImageNet (Le & Yang, 2015) to CIFAR-10 (Krizhevsky et al., 2009) instead of contextual shifted far-OOD data like MNIST (LeCun et al., 1998) to CIFAR-10.

To validate the assumptions made in Section 4 for AL, we analyze the latent space configuration in our AL experiments. The objective of our method is to select samples in the decision boundary region for the AL case. In Fig. 5, we compare CoreSet and Loss Learning with SISOM. CoreSet shows high diversity in unseparated regions. The pseudo-uncertainty-based Loss Learning is concentrated in its selection but fails to diversify the selection across all decision boundaries. In contrast, SISOM, as depicted in Fig. 5c, effectively targets the decision boundary and covers the area between unseparated samples, demonstrating its effectiveness for AL and near-OOD.

**Active Learning:** AL exists in numerous variants; given the focus on showing the relation between OOD detection and AL, we adhered to common close-set AL benchmark settings (Yoo & Kweon, 2019; Ash et al., 2020) with the in this context widely evaluated datasets, namely CIFAR-10 and CIFAR-100 (Krizhevsky et al., 2009), as well as SVHN (Netzer et al., 2011), in conjunction with different ResNet models (He et al., 2016). Since we assume a complete learning life-cycle, we mostly focus on the high and mid data regions (Lüth et al., 2023), as low data regions would not allow reasonable performance for deployment. We selected our query sizes accordingly and followed the commonly suggested sizes of $q = 1000$ for CIFAR-10 (Yoo & Kweon, 2019; Lüth et al., 2023) and $q = 2000$ for CIFAR-100 (Caramalau et al., 2021). We employ several baselines, including **CoreSet** (Sener & Savarese, 2018), **CoreGCN** (Caramalau et al., 2021), **Random**, **Badge** (Ash et al., 2020), and **Loss Learning (LLoss)** (Yoo & Kweon, 2019). Additionally, we adapted **NAC** (Liu et al., 2024) from OOD detection to AL to assess the transferability from OOD to AL.

Since more recent AL methods use pre-text tasks or unsupervised weights to select samples, we further consider an additional Semi-Supervised AL setup for a fair comparison, including more recent approaches. In this setting we compare SISOM with **TypiClust** (Hacohen et al., 2022), **ProbCover** (Yehuda et al., 2022) and **PT4AL** (Yi et al., 2022). Additionally, in this setup, SISOM can profit from the well-defined feature space. All experiment details, including parameters and settings, are in Appendix B.1.

CIFAR-10: In the CIFAR-10 benchmark depicted in Fig. 6a, SISOM and SISOMe demonstrate rapid progress and consistent performance, surpassing other methods in all selection cycles. Furthermore, as the sample size increases, our method maintains its superiority over Learning Loss and CoreSet. NAC does not demonstrate superior performance compared to Random. To underline the versatility of SISOM , we analyze additional mid-data regimens and different query sizes as proposed by Lüth et al. (2023) in Appendix A.3. In the mid data regime in Fig. 10a. In this data regime, SISOM and SISOMe can maintain their performance shown in the high data region. As SISOM does not involve continuous distance updates and applies a top-k selection, it could suffer from less diversity in the selected batch. To investigate this problem, we perform additional experiments with reduced and increased query sizes in Fig. 9a and Fig. 9b (Appendix A.2), respectively. SISOM 's high performance in both settings strengthens our hypothesis that targeting the areas between clusters leads to higher batch diversification than purely uncertainty-based approaches.

CIFAR-100: After examining SISOM in datasets with a limited number of classes, we examine the AL setup on the larger CIFAR-100 dataset and report the results in Fig. 6b. In this setting, all methods are less stable in their ranking compared to the other dataset, reflecting the increased difficulty of the dataset. The complexity of the dataset requires more data for the model to perform effectively. While in the early stages, pure diversity-based methods are in the lead, SISOM gains velocity in the last selection steps and achieves the highest performance difference only in the last step SISOMe is more effective.

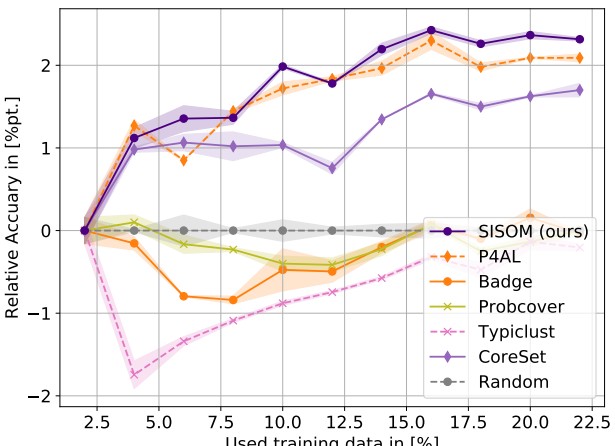

Figure 7: Comparison of accuracy relative to random selection of active learning methods on a semi-supervised CIFAR-10 setup with indicated standard errors for ResNet 18.

Semi-Supervised: In Fig. 7, we compare different approaches designed for a semi-supervised AL setup. For this experimental setup, every approach builds upon the unsupervised pre-training weights, which are required for TypiClust and ProbCover. While both approaches do not perform strongly in the classic data regime, PT4AL shows a good performance by leveraging training on auxiliary pre-tasks. However, SISOM strongly profits from the pre-training and outperforms PT4AL *without* additional pre-task training.

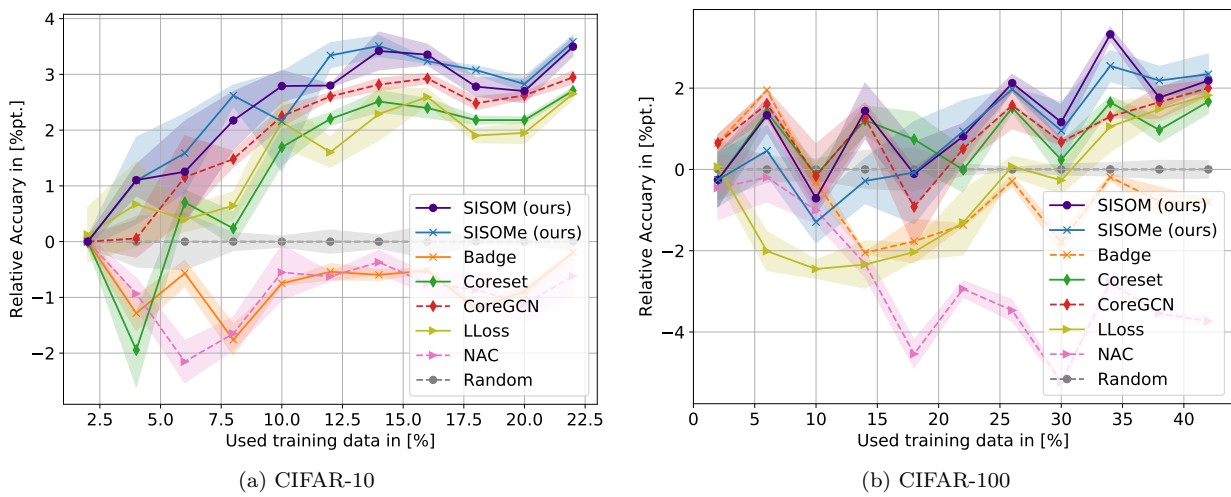

(a) CIFAR-10          (b) CIFAR-100

Figure 6: Comparison of different active learning methods on CIFAR-10 and CIFAR-100 with ResNet18 with indicated standard errors. Relative accuracy is the absolute performance difference between a method and random selection.

SVHN: Following the experiments on CIFAR-10 and CIFAR-100 we conduct experiments on SVHN and report them in Appendix A.1.

Additionally Active Learning Experiments: To underline the flexibility of SISOM, we conduct additional experiments on different models. Appendix A.3 and Fig. 10b show experiments with a larger ResNet version. The larger model leads to lower performance differences between all methods. SISOMe struggles in the early cycles with latent space assessment but eclipses others and SISOM after a few cycles. Finally, SISOM and SISOMe achieve the highest accuracy in the final cycles.

In all AL experiments, SISOM achieved state-of-the-art performance across all three datasets, confirming its viability for AL. Although SISOM initially lags behind on CIFAR-100, it surpasses other methods in later selection cycles as more training data improves feature space separation.

**Out-of-Distribution Detection:** Following the classic AL benchmarks, we utilize the OpenOOD framework (Yang et al., 2022; Zhang et al., 2023) to evaluate on the OOD detection task. We employ the recommended benchmarks on CIFAR-10 (Krizhevsky et al., 2009), CIFAR-100 (Krizhevsky et al., 2009), and ImageNet 1k (Deng et al., 2009) following the framework's OOD datasets categorization (Yang et al., 2022; Zhang et al., 2023) (provided in Appendix B.2). For a fair evaluation, we use the official leader board, Cross-Entropy training checkpoints, and report near- and far-OOD results. The results are reported as aligned to the framework without standard deviation. In addition, we evaluate the life cycle setting. For all experiments, we employ the implementation provided by the OpenOOD framework when it is available. The baselines used for validation include **NAC** (Liu et al., 2024), **Ash** (Djurisic et al., 2022), **KNN** (Sun et al., 2022), **Odin** (Hsu et al., 2020), **ReAct** (Sun et al., 2021), **MSP** (Hendrycks & Gimpel, 2016), **Energy** (Liu et al., 2020), **Dice** (Sun & Li, 2022), **SCALE** (Xu et al., 2024), **CombOOD** (Rajasekaran et al., 2024), **NNGuide** (Park et al., 2023a), **RankFeat** (Yu et al., 2023), **FeatureNorm** (Song et al., 2022) and **GEN** (Liu et al., 2023) and were selected based on their near-OOD performance. Moreover, we ported the **CoreSet** (Sener & Savarese, 2018) AL method to verify the transferability from AL to OOD.

In Table 1, we examine the performance of SISOM and SISOMe for the three benchmarks. As stated before, the near-OOD data is more relevant for this task. We focus on the near-OOD evaluation and report the far-OOD results in Appendix A.5. We rank the methods by their combined rank over the three benchmarks (Appendix B.2). For the *CIFAR-10* benchmark, SISOMe and SISOM achieve the highest AUROC score for near-OOD data, respectively. In the *CIFAR-100* evaluation, SISOMe ranks as the third-best method for near-OOD and stands apart from the individual metrics, SISOM and Energy. GEN, which shows weaker performance in the other setting, achieves first place. This is an interesting finding since, in contrast to CIFAR-10, Energy achieves better performance than SISOM among the individual metrics on CIFAR-100. This supports our hypothesis that we obtain stronger performances in well-separated and poorly-separated feature spaces by considering the average ratio $r_{avg}$ as a proxy for feature separation. The *ImageNet 1k* benchmark suggested by OpenOOD contains more classes and is a much larger dataset than the previous ones. In this setup SISOMe achieves the third-best scores on near-OOD, making it the only method with three top-three rankings. In addition, when aggregating the individual ranks across the three benchmarks,

Table 1: Near-OOD AUROC benchmark for CIFAR-10, CIFAR-100, and ImageNet1k with Cross-Entropy training. Dataset and official checkpoints according to OpenOOD. It can be seen that most methods focus on one particular setting. Given the number of baselines for each dataset, we estimated a rank for each benchmark and aggregated them for an overall ranking reported in the first row. The top three ranks are marked in gray from dark to light, with the top method name also in bold. For individual datasets, the top three ranks are indicated from first to third using **bold**, double underlined or single underlined formatting.

| Dataset | SISOMe* | SISOM | NAC | KNN | CoreSet | GEN | MSP | Energy | ReAct | Feat.Norm | ODIN | RankFeat | KLM | DICE | ASH | SCALE | CombOOD | NNGuide |
|---|---|---|---|---|---|---|---|---|---|---|---|---|---|---|---|---|---|---|
| Overall Rank | **1** | 4 | 10 | 9 | 14 | 2 | 8 | 7 | 6 | 16 | 11 | 17 | 13 | 15 | 12 | 5 | 3 | 8 |
| CIFAR-10 | **91.76** | 91.40 | 90.93 | 90.64 | 90.34 | 88.20 | 88.03 | 87.58 | 87.11 | 85.52 | 82.87 | 79.46 | 79.19 | 78.34 | 75.27 | 82.55 | 91.13 | 87.56 |
| CIFAR-100 | 81.10 | 79.42 | 75.90 | 80.18 | 75.69 | **81.31** | 80.27 | 80.91 | 80.77 | 47.87 | 79.90 | 61.88 | 76.56 | 79.38 | 78.20 | 80.99 | 78.77 | 81.25 |
| ImageNet 1k | 78.59 | 77.33 | 74.43 | 71.10 | - | 76.85 | 76.02 | 76.03 | 77.38 | 67.57 | 74.75 | 50.99 | 76.64 | 73.07 | 78.17 | 81.36 | **95.22** | 73.57 |

Table 2: OOD benchmark for CIFAR-10 using the AL checkpoints of SISOM .

| AUROC | SISOMe | ReAct | GEN | MSP | ASH | NAC | RankFeat |
|---|---|---|---|---|---|---|---|
| near-OOD | **86.84** | **86.84** | 85.43 | 84.37 | 83.39 | 82.26 | 60.20 |
| far-OOD | **88.39** | 87.72 | 86.04 | 84.85 | 87.33 | 85.06 | 56.73 |

SISOMe ranks first. Interestingly, while NAC showed high performance in CIFAR-10, it ranks much lower, and KNN, the third-best method in CIFAR-10, ranks last. Meanwhile, ASH, which ranks forth in this benchmark, ranked last in the CIFAR-10 setting. This underlines the difficulty of the different datasets and their benchmarked OOD pairs. In the real-world applications, less relevant *far-OOD* evaluation in Appendix A.5, the dependence on the performance between datasets and methods persists. Despite this, SISOMe and SISOM achieve high scores, with SISOMe remaining the only method with a top-3 ranking in all benchmarks, leading to an overall top-one rank.

**Full Life Cycle:** To evaluate the effectiveness of SISOM in a life cycle setting, we conduct OOD benchmark experiments with models trained in an AL cycle. We use the same setting as for the benchmark CIFAR-10 experiments with similar near- and far-OOD. The exact setup and training scheme of the checkpoints is described in Appendix B. In Table 2, all methods suffer from less training data. However, SISOMe achieves the top performance for both OOD categories, making it suitable for the full application life cycle. A more detailed analysis, including different intermediate cycles, is shown in Appendix A.6.

**Overall benchmarks**, SISOMe is the only approach, being consistently under the top three ranks and secured first place in the overall ranking. Excluding SISOMe , SISOM achieved one top-one ranking. As intended, our method performs relatively better on near-OOD data than on far-OOD data. This is understandable, as the ratio between inner and outer class distance is higher for data close to the training data distribution, while the quotient is lower for far-OOD. Additionally, near-OOD is closer to the data of interest for AL selection. According to Yang et al. (2022), near-OOD is considered the more challenging task and is more likely to occur in real-world applications. Thus, a higher score on near-OOD may be preferred in practice.

**Ablation Studies:** In an ablation study, we examine the effect of unsupervised feature space analysis and reduce the labeled set $\mathbb{T}$. A study of the individual components of SISOM is given in Appendix A.4.

Optimal Sigmoid Steepness: In our feature space analysis in Section 4, we derived $r_{\text{avg}}$ in Eq. (10) as a proxy for the feature space separability. Due to the distance concept of SISOM , we hypothesize that it works better in well-separated feature spaces. To examine this, we conduct a random search for different $\alpha$ sets and record the different $r_{\text{avg}}$ values. To reduce the search space, we follow the premise postulated in Section 4 that generally, deeper layers require a steeper sigmoid curve, i.e., a higher $\alpha_j$ value due to the nature of the features captured within these layers. After computing every $r_{\text{avg}}$ value for each combination of $\alpha$, we select the $\alpha_{\text{opt}}$ set that minimizes $r_{\text{avg}}$. Formally, this can be written as:

$$\alpha_{\text{opt}} = \arg\min_{\alpha} r_{\text{avg}}(\alpha)$$

In Table 3, an optimized set $\alpha_{opt}$ is marked with OS. As it can be seen, a set with better feature space separation leads to increased performance for CIFAR-100 and ImageNet, partly confirming our hypothesis. In CIFAR-10, however, the original set of parameters yields the best results. One explanation might be that, in CIFAR-10, the different classes are already well separated, such that optimization on this separation yields no improvement and leads to an overfitting behavior.

Reduced Subset Selection: For larger datasets, distance-based approaches like CoreSet (Sener & Savarese, 2018) and Badge (Ash et al., 2020) face significant computational challenges, impacting OOD detection. In Section 4, we proposed using a reduced subset $\mathbb{T}$ of the comparison set $\mathbb{Z}_L$. For each dataset, we select a total of 10% of the samples for each class, drastically increasing inference speed. The impact of our reduced subset selection (RS) is compared in Table 3 and illustrated in Fig. 4d. The preprocessing steps for SISOM in Table 3 show improved AUROC near-OOD scores for all datasets, supporting our earlier hypothesis. While feature analysis and pre-selection enhanced performance for ImageNet and CIFAR-100, CIFAR-10 did not

Table 3: Ablation study on Optimal Sigmoid Steepness (OS) and Reduced Subset Selection (RS) on near OOD benchmarks.

| | ImageNet | | CIFAR 100 | | CIFAR 10 | |
|---|---|---|---|---|---|---|
| Method | $AUROC_n$ | $r_{avg}$ | $AUROC_n$ | $r_{avg}$ | $AUROC_n$ | $r_{avg}$ |
| SISOM | 77.21 | 0.270 | 75.93 | 0.33 | 91.33 | 0.26 |
| SISOM , OS | **77.4** | 0.266 | 79.56 | 0.19 | 90.37 | 0.099 |
| SISOM , RS | 77.33 | 0.249 | 76.07 | 0.31 | **91.40** | 0.24 |
| SISOM , OS, RS | 77.37 | 0.245 | **79.69** | 0.18 | 90.54 | 0.086 |

Table 4: Runtime comparison of different subset selection sizes for SISOMe in seconds. The setup time is the time to build up the set for comparison, while eval time measures one run over all OpenOOD evaluation datasets for CIFAR-100.

| Subset Size $\mathbb{T}$ | 100% | 60% | 20% | 5% |
|---|---|---|---|---|
| Setup Time [s] | $291.17 \pm 1.14$ | $60.87 \pm 0.93$ | $40.63 \pm 0.79$ | $14.53 \pm 0.92$ |
| Eval Time [s] | $885.67 \pm 3.31$ | $688.34 \pm 6.27$ | $376.57 \pm 1.60$ | $251.67 \pm 0.74$ |
| AUROC near OOD | 80.97 | 81.01 | 81.03 | 81.10 |

show similar improvements. Considering the low average relevance score, the selected values may have overly constrained the feature space, resulting in an overfitting behavior. Further details in Appendix B.3.

Runtime Considerations: In Table 4, we compare the runtime of SISOMe for different subset sizes for a full OpenOOD CIFAR-100 evaluation. It can be seen that with decreasing subset size, the computational efficiency increases for setup and evaluation time. Perhaps surprisingly, even the performance is slightly improving with decreasing subset size. One reason might be that a smaller subset is a more compact, outlier-free representation of the data distribution. We also compare SISOM 's and SISOMe 's runtime to other OOD detection and AL methods in Appendix A.6. In general, approaches without distance calculations are faster than diversity-based methods such as SISOM .

## 6 Conclusion

We proposed SISOM , the first approach designed explicitly to solve OOD detection and AL jointly, providing an effective simplification in real-world application life cycles by eliminating an additional OOD detection design phase and avoiding conflicting goals of AL and OOD detection. By weighting latent space features with KL divergence of the neuron activations and relating them to the latent space clusters of the different classes SISOM achieves state-of-the-art performance in both tasks, *without* requiring a specific training scheme. In addition, SISOM provides a novel feature space analysis scheme enabling a post-training feature space refinement as well as a self-balancing uncertainty and diversity fusion introduced as SISOMe . Our experiments show that unlabeled samples and near-OOD data can be ambiguous, which SISOM can leverage. In the common OpenOOD benchmarks, SISOM achieves the *top-1* performance in *one of the three benchmarks* and is the only method with top-three places in all benchmarks for challenging near-OOD scenario. For AL, SISOM surpasses state-of-the-art approaches in eight different benchmark settings. While current state-of-the-art approaches are highly specialized for either AL or OOD detection, SISOM solves both tasks with the same approach. This versatility makes SISOM well-suited for real-world applications, like environment sensing, which face high label costs, abundant unlabeled data, and OOD samples at inference. In addition, SISOM provides important insights into the ambiguity of unlabeled and near-OOD samples.

**In future work**, we aim to leverage the relation of near-OOD data and unlabeled samples to explore tasks setup like open-set AL where currently two components are required. In addition, we aim extend SISOM to more complex tasks and investigate the effect batch diversification techniques on our two task approach.

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

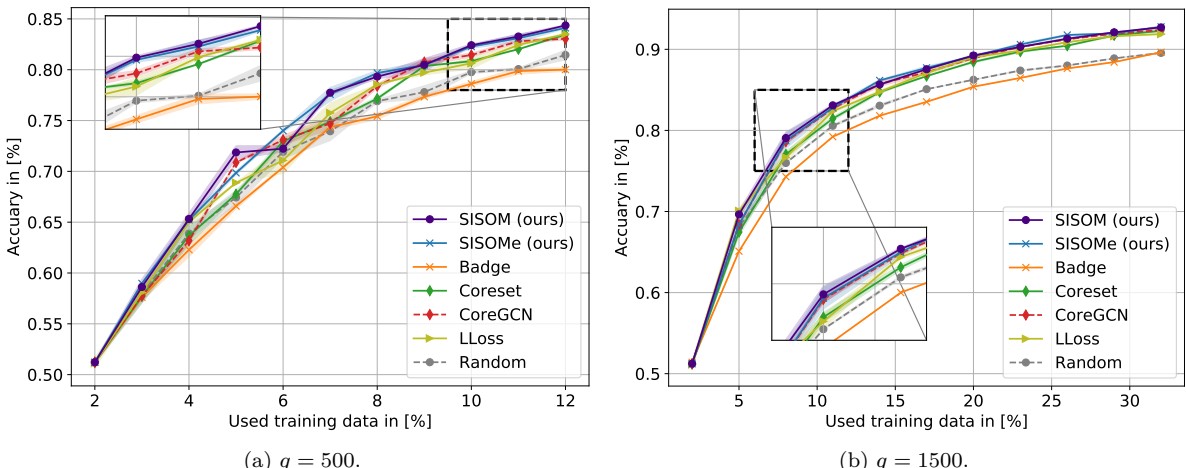

(a) $q = 500$.

(b) $q = 1500$.

Figure 9: Comparison of different query sizes for CIFAR-10 with ResNet18. It can be seen, that SISOM and SISOMe are quite robust for a variation of the selection size.

# A   Additional Experiments

In this section, we present additional experiments for SISOM.

## A.1   Active Learning - SVHN

Following the CIFAR experiments settings, we depict the results for the SVHN experiments in Fig. 8 with a query size of $q = 500$. Similar to the CIFAR-10 results, our method maintained high performance, but the method differences shrink with the easier the dataset. In the last cycle, SISOM reaches the highest performance, with a margin over other methods. As for CIFAR-10, NAC did not perform well in the data selection. Given that SVHN's 10 classes are numbers, it is easier than the more diverse CIFAR-10 benchmark dataset. This can be observed by an overall reduced performance gap between the methods compared to CIFAR-10. LLoss starts with a strong performance, which can be explained by the additional loss of learning loss and network extensions.

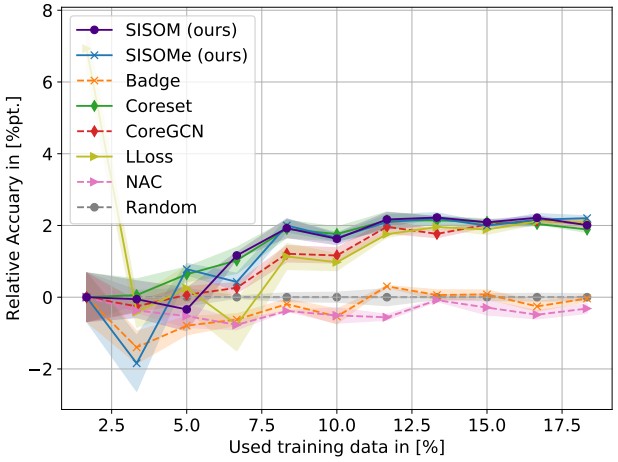

Figure 8: Comparison of accuracy relative to random selection for different active learning methods on SVHN with indicated standard errors.

## A.2   Query Size Analysis

The query size is an important design factor of AL experiments with a measurable impact on the performance of methods Lüth et al. (2023). To investigate how large and small query sizes impact the performance of SISOM, we conduct experiments with $q = 500$ and $q = 1500$. In Fig. 9a, with a small query size, SISOM and SISOMe can increase their performance lead compared to other methods. For a larger query size (Fig. 9b), the differences shrink as the larger selection size leads to faster model convergence.

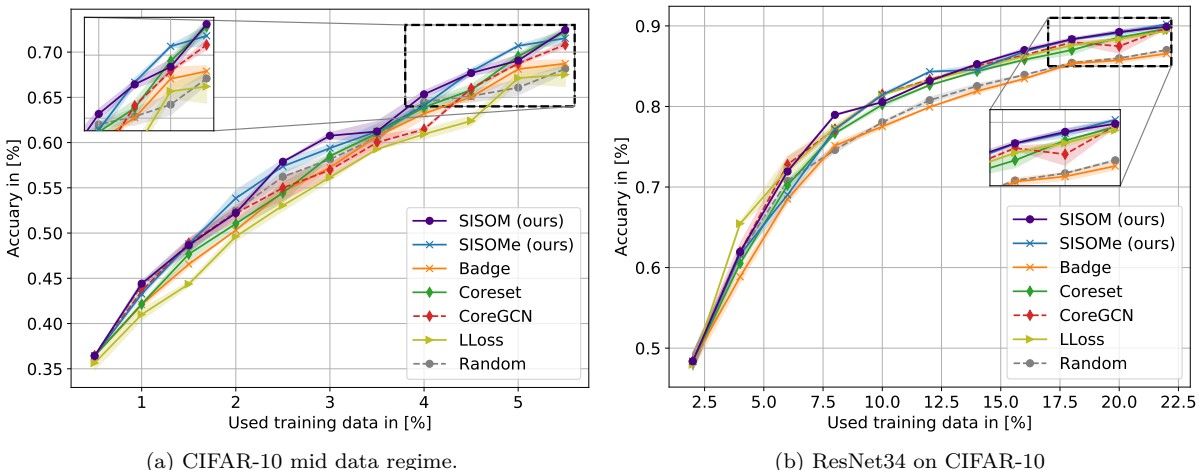

(a) CIFAR-10 mid data regime.

(b) ResNet34 on CIFAR-10

Figure 10: Additional experiments for active learning, with a) CIFAR-10 in the mid range data regime and b) ResNet34 for high data regime aligned with our experiments in Section 5.

### A.3 Further Active Learning Experiments

**CIFAR-10 Medium Data Region:** In addition to our primary high data region experiments in the main paper we conduct experiments in the medium data region in Fig. 10a. In this setting, the initial set size and the query size are set to 250, as recommended by Lüth et al. (2023) It can be seen that SISOM can keep its good performance in this scenario as well, while in the early phase, the selection of SISOM targets the right samples. The less defined latent space lets it leak behind SISOMe in cycle 2. In general the performance differences are more inconsistent compared to the high data region.

**Additional Models:** In addition to our experiments in the main paper, we verify the effectiveness of SISOM for different models. In Fig. 10b, we see the performance of all methods converges fast given the larger model. However, in the last cycles, both SISOM and SISOMe remain on the top of the ranking.

### A.4 Feature Space Assignments

In this section, we highlight the influence of major components of our methods on the ability to separate InD and OOD data. In Fig. 11, we display the influence of the KL divergence gradient with a T-SNE analysis on CIFAR-10 (Krizhevsky et al., 2009) as InD and Tiny ImageNet (tin) (Le & Yang, 2015) as near-OOD. Without feature enhancement, the latent space is much harder to separate, and tin is distributed all over the latent space as shown in Fig. 11a. In contrast, the latent space with KL divergence enhances features, is much more separated, and has a clearer decision boundary to the near classes as indicated in Fig. 11b.

In addition to the previously presented density plots, we show the inner and outer distance together with the distance quotient of SISOM in Fig. 12 for CIFAR-10. Fig. 12a shows the inner class, indicating small inner class distances leading to a good separability for the InD data. On the other hand, the outer class distance in Fig. 12b provides a good separable peak for InD data, but a portion of InD overlaps with OOD data. The combined distance quotient shows the increased separability of the different InD and OOD sets as depicted in Fig. 12c.

### A.5 Extended OOD Benchmark Results

In Table 5, we present the results of far-OOD along with near-OOD results of the three benchmarks CIFAR-10, CIFAR-100, and ImageNet 1k. It can be seen that SISOMe achieves for CIFAR-10 top-1 position for near- and far-OOD for CIFAR-10. For CIFAR-100 and ImageNet 1k SISOMe achieves the highest near-OOD score but shows weaker scores for far-OOD. This can be explained by the additional focus on AL, which is more related to near-OOD. Overall, it is apparent that most methods are primarily focused on either

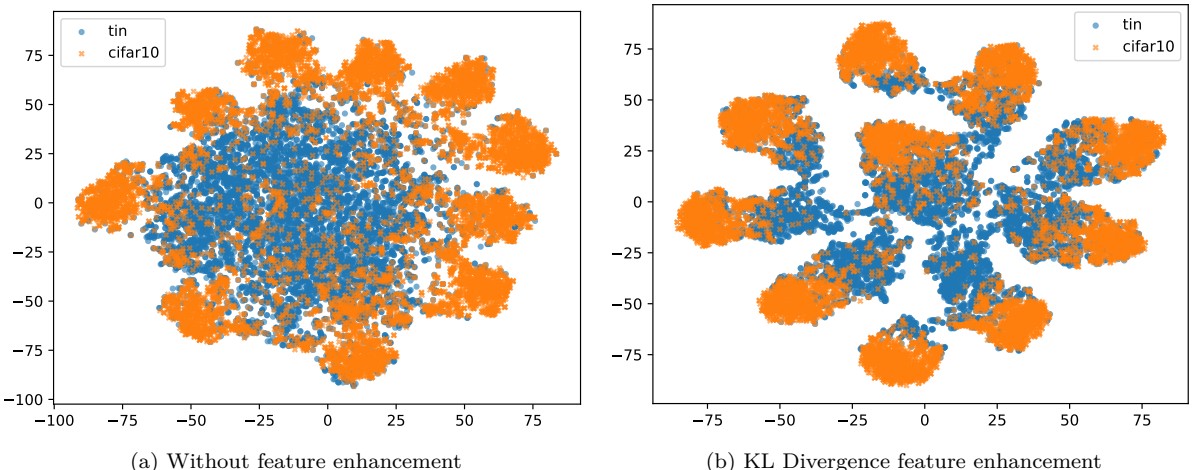

(a) Without feature enhancement

(b) KL Divergence feature enhancement

Figure 11: T-SNE comparison of the latent space for OOD detection with and without KL-Divergence feature enrichment.

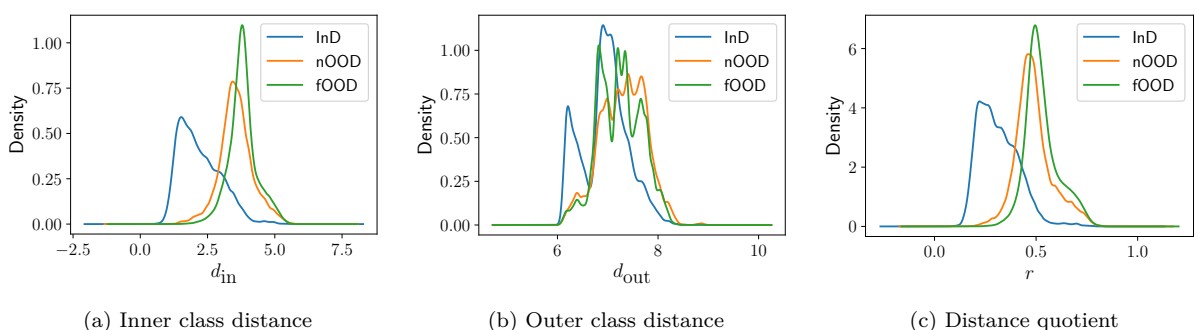

(a) Inner class distance

(b) Outer class distance

(c) Distance quotient

Figure 12: Density plots for the inner class distance, outer class distance, and the distance quotient of SISOM for CIFAR-10 with near-OOD (nOOD) and far-OOD (fOOD) as defined in OpenOOD.

near-OOD or far-OOD, creating a trade-off between these two metrics. Additionally, the performance of individual methods varies significantly across different datasets. For instance, GEN ranks first in near-OOD for CIFAR-100 but is outperformed by SISOMe in the far-OOD category. Furthermore, for CIFAR-10 and ImageNet 1k, it has not even a top-3 rank for near- or far-OOD. In this context, the results of SISOMe with 4 top-3, including one top-1 position are outstanding, especially in the targeted near-OOD setting. Details on the rank calculations are given in Appendix B.2.

### A.6 Extended Life Cycle Experiments

In Fig. 13, we evaluated the life cycle performance when deployed for OOD detection as shown in Fig. 3. For near-OOD SISOMe is always the best method, while for far-OOD SISOMe shows higher performance in higher cycles. Since near-OOD is closer to AL, a relatively higher performance is preferred in practice. As for Table 2, we used AL checkpoints of SISOMe for the experiments.

Table 5: far-OOD (f) and near-OOD (n) benchmark for CIFAR-10, CIFAR-100, and ImageNet 1k with Cross-Entropy training. Dataset and official checkpoints according to OpenOOD Yang et al. (2022) benchmark setting for the respective datasets. Given the number of baselines, for each dataset, we estimated a rank for each category and calculated the joint ranking over all datasets for near-OOD and far-OOD. For individual datasets, the top three ranks are indicated from first to third using **bold**, double underlined or single underlined formatting.

| Dataset | | SISOMe* | SISOM | NAC | KNN | CoreSet | GEN | MSP | Energy | ReAct | Feat.Norm | ODIN | RankFeat | KLM | DICE | ASH | SCALE | CombOOD | NNGuide |
|---|---|---|---|---|---|---|---|---|---|---|---|---|---|---|---|---|---|---|---|
| CIFAR-10 | n | **91.76** | 91.40 | 90.93 | 90.64 | 90.34 | 88.20 | 88.03 | 87.58 | 87.11 | 85.52 | 82.87 | 79.46 | 79.19 | 78.34 | 75.27 | 82.55 | 91.13 | 87.56 |
| | f | 94.74 | 94.50 | 94.60 | 92.96 | 92.85 | 91.35 | 90.73 | 91.21 | 90.42 | **95.59** | 87.96 | 75.87 | 82.68 | 84.23 | 78.49 | 86.39 | 94.65 | 90.88 |
| CIFAR-100 | n | 81.10 | 79.42 | 75.90 | 80.18 | 75.69 | **81.31** | 80.27 | 80.91 | 80.77 | 47.87 | 79.90 | 61.88 | 76.56 | 79.38 | 78.20 | 80.99 | 78.77 | 81.25 |
| | f | 80.16 | 77.91 | **86.98** | 82.40 | 79.53 | 79.68 | 77.76 | 79.77 | 80.39 | 80.99 | 79.28 | 67.10 | 76.24 | 80.01 | 80.58 | 81.42 | 85.87 | 81.21 |
| ImageNet 1k | n | 78.59 | 77.33 | 74.43 | 71.10 | - | 76.85 | 76.02 | 76.03 | 77.38 | 67.57 | 74.75 | 50.99 | 76.64 | 73.07 | 78.17 | 81.36 | **95.22** | 73.57 |
| | f | 89.04 | 88.01 | 95.29 | 90.18 | - | 89.76 | 85.23 | 89.50 | 93.67 | 91.13 | 89.47 | 53.93 | 87.60 | 90.95 | 95.74 | **96.53** | 90.24 | 93.82 |
| Near Rank | | **1** | 4 | 10 | 9 | 14 | 2 | 8 | 7 | 6 | 16 | 11 | 17 | 13 | 15 | 12 | 5 | 3 | 8 |
| Far Rank | | 7 | 13 | **1** | 4 | 14 | 10 | 16 | 11 | 8 | 2 | 15 | 18 | 17 | 12 | 9 | 6 | 3 | 5 |

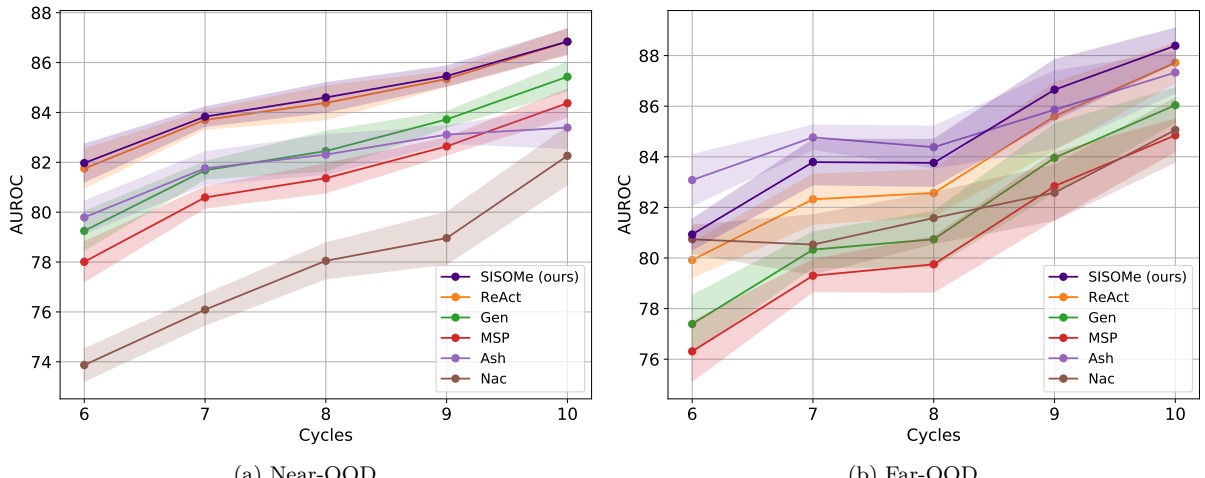

(a) Near-OOD

(b) Far-OOD

Figure 13: Full life cycle experiments evaluating near-OOD and far-OOD AUROC for different active learning checkpoints.

## A.7 Time Analysis

**Analysis on OOD Detection:** Table 6 outlines the runtime for various OOD detection methods on CIFAR-100. Our method, SISOMe, recorded a total time of $1177.50 \pm 3.27$ seconds, with the evaluation phase comprising $885.66 \pm 3.31$s for the full set evaluation. This runtime is longer than other evaluated methods due to the computationally expensive pairwise distance calculations in SISOMe. However, with our reduced subset to 5%, we can significantly reduce the runtime by almost 400%, which brings SISOMe in the same range as other distance-based approaches like KNN or NNGuide.

**Analysis on Active Learning:** The per-cycle runtime analysis for AL methods on CIFAR-100 is presented in Table 7. Our proposed methods, SISOM and SISOMe, demonstrate runtimes substantially more efficient than the diversity-based Badge method (e.g., $33664 \pm 6682$s in Cycle 0) and are competitive with, or often faster than, Coreset. However, methods that leverage graph neural networks for dimensionality reduction, like CoreGCN, or individual evaluation strategies without distance calculations, such as LLoss and NAC, naturally exhibit faster per-cycle execution times due to their less complex selection mechanisms.

Table 6: OOD Runtime Analysis for different methods with CIFAR-100 as the InD dataset. Time is measured on a desktop PC with A6000 and 64GB of RAM. Total time is the sum of setup time and eval time.

| | SISOMe (ours) 100% | SISOMe (ours) 5% | NAC | KNN | GEN | MSP | ReAct | ODIN | NNGuide |
|---|---|---|---|---|---|---|---|---|---|
| Total Time (s) | 1177.50 $pm$ 3.27 | 266.88 ± 1.19 | 63.5 ± 1.4 | 188.4 ± 11.2 | 23.8 ± 1.4 | 23.8 ± 1.5 | 31.8 ± 1.1 | 73.7 ± 1.3 | 270.5 ± 4.2 |
| Eval Time (s) | 885.66 ± 3.31 | 251.67 ± 0.74 | 57.7 ± 0.4 | 166.2 ± 11.4 | 22.3 ± 0.7 | 22.3 ± 0.7 | 22.9 ± 0.4 | 55.1 ± 0.1 | 259.4 ± 0.2 |

Table 7: Active Learning runtime analysis for methods with CIFAR-100 in the high data selection regime. Time is measured on a Kubernetes pod with V100 and 24GB of RAM from a Nvidia DGX machine. *CoreCGN leverages a graph neural network to reduce the dimensionality before calculating the distances, which grands more efficient distance calculations.

| | Diversity-based | | | | Individual Evaluation | | |
|---|---|---|---|---|---|---|---|
| | SISOM (ours) | SISOMe (ours) | Badge | Coreset | CoreGCN* | LLoss | NAC | Random |
| Cycle 0 | 1477 ± 896 | 954 ± 126 | 33664 ± 6682 | 2604 ± 1572 | 529 ± 121 | 7.46 ± 0.19 | 25.50 ± 0.18 | 0.12 ± 0.01 |
| Cycle 1 | 1492 ± 865 | 1084 ± 130 | 31019 ± 5752 | 2740 ± 1753 | 502 ± 115 | 5.81 ± 0.05 | 25.06 ± 0.04 | 0.01 ± 0.00 |
| Cycle 2 | 1477 ± 782 | 1117 ± 127 | 34229 ± 9662 | 2886 ± 1612 | 529 ± 111 | 5.43 ± 0.05 | 25.20 ± 0.12 | 0.01 ± 0.00 |
| Cycle 3 | 1374 ± 693 | 1077 ± 165 | 27274 ± 3331 | 2203 ± 866 | 511 ± 131 | 5.21 ± 0.02 | 25.63 ± 0.17 | 0.01 ± 0.00 |
| Cycle 4 | 1432 ± 744 | 1124 ± 131 | 25676 ± 1280 | 2213 ± 889 | 519 ± 114 | 4.91 ± 0.06 | 25.83 ± 0.36 | 0.01 ± 0.00 |
| Cycle 5 | 1445 ± 566 | 1178 ± 91 | 27286 ± 4139 | 2030 ± 900 | 506 ± 114 | 4.65 ± 0.05 | 25.69 ± 0.06 | 0.01 ± 0.00 |
| Cycle 6 | 1372 ± 654 | 1176 ± 93 | 23590 ± 3793 | 1780 ± 741 | 520 ± 112 | 4.46 ± 0.01 | 25.88 ± 0.07 | 0.01 ± 0.00 |
| Cycle 7 | 1160 ± 410 | 1150 ± 72 | 20324 ± 1041 | 2172 ± 1177 | 535 ± 86 | 4.19 ± 0.09 | 26.22 ± 0.13 | 0.01 ± 0.00 |
| Cycle 8 | 1151 ± 397 | 1121 ± 70 | 19328 ± 284 | 1704 ± 818 | 510 ± 113 | 3.75 ± 0.01 | 26.63 ± 0.09 | 0.01 ± 0.00 |
| Cycle 9 | 1197 ± 474 | 1126 ± 28 | 18532 ± 526 | 1529 ± 622 | 500 ± 133 | 3.59 ± 0.04 | 27.02 ± 0.14 | 0.01 ± 0.00 |

# B  Experimental Details

In this section, we provide experiment details to support the reproducibility of results by providing the used parameters.

## B.1  Active Learning Experiments

In AL experiments, we used a ResNet18 and ResNet34 (He et al., 2016) model, with the suggested modifications of Yoo & Kweon (2019) presented in a CIFAR benchmark repository (kuangliu, 2021), which replaced the kernel of the first convolution with a $3 \times 3$ kernel. Additionally, we used an SGD optimizer with a learning rate of 0.1 and multistep scheduling at 60, 120, and 160, decreasing the learning rate by a factor of 10, which are reported benchmark parameters for CIFAR-100 (weiaicunzai, 2022). For SVHN and CIFAR-10 we used a learning rate of 0.025 and a cosine scheduler as suggested by Yehuda et al. (2022). For the construction of the feature space, we used the layers after the 4 blocks of ResNet with sigmoid steepness parameters as reported in Table 8.

Table 8: Sigmoid steepness parameters for active learning experiments.

| Dataset | Adaptive Average Pooling Layer | Sequential Layer 3 | Sequential Layer 2 | Sequential Layer 1 |
|---|---|---|---|---|
| CIFAR-10 | 50 | 10 | 1 | 0.05 |
| CIFAR-100/SVHN | 1 | 0.1 | 0.1 | 0.1 |

For the Semi-Supervised experiments, we followed the suggestion of Yehuda et al. (2022) and used a model pre-train but used the SIMCLR (Chen et al., 2020) pre-training checkpoint instead of the SCAN (Gansbeke et al., 2020) approach. While Yehuda et al. (2022) used the weights only for the selection and trained a supervised model from scratch, we also initialized the task models with this checkpoint.

## B.2 Out-of-Distribution Experiments

In the OOD experiments, we report the mean of the three different seeds employed in the standard setting of the OpenOOD[2] (Yang et al., 2022) framework with ResNet18 for CIFAR-10 and CIFAR-100. For Imagenet, we use the sole ResNet50 torchvision checkpoint provided in the standard settings. We utilized the near- and far-OOD assignments suggested by the benchmark listed below. We followed the official tables of OpenOOD's benchmark and reported the mean without the standard deviation. For the CIFAR-100 experiment, instead of using the automated $r_{\text{avg}}$ value to balance between $r$ and $E$ from Eq. (11), we set $r_{\text{avg}} = 0.8$ for SISOMe based on a hyperparameter study. In the benchmark tables, we reported for SISOM the best values matching the best values of the ablation study modifications. Furthermore, we follow the suggested sigmoid steepness parameters (Liu et al., 2024) for CIFAR-10 and ImageNet. For CIFAR-100, we choose values that minimize $r_{avg}$. A detailed overview of the sigmoid steepness parameters for the 4 blocks of ResNet18 and ResNet50 for all experiments is provided in Table 9:

Table 9: Sigmoid steepness parameters for OOD Detection experiments.

| Dataset | Adaptive Average Pooling Layer | Sequential Layer 3 | Sequential Layer 2 | Sequential Layer 1 |
|---|---|---|---|---|
| CIFAR-10 | 100 | 1000 | 0.001 | 0.001 |
| CIFAR-100 | 1 | 0.1 | 0.1 | 0.1 |
| ImageNet | 3000 | 300 | 0.01 | 1 |

OOD dataset assignment according to OpenOOD (Yang et al., 2022):

- **CIFAR-10**
  Near-OOD: CIFAR-100 (Krizhevsky et al., 2009), Tiny ImageNet (Le & Yang, 2015)
  Far-OOD: MNIST (LeCun et al., 1998), SVHN (Netzer et al., 2011), Textures (Cimpoi et al., 2014), Places365 (López-Cifuentes et al., 2020)

- **CIFAR-100**
  Near-OOD: CIFAR-10 (Krizhevsky et al., 2009), Tiny ImageNet (Le & Yang, 2015)
  Far-OOD: MNIST (LeCun et al., 1998), SVHN (Netzer et al., 2011), Textures (Cimpoi et al., 2014), Places365 (López-Cifuentes et al., 2020)

- **ImageNet**
  Near-OOD: SSB-hard (Vaze et al., 2021), NINCO (Bitterwolf et al., 2023)
  Far-OOD: iNaturalist (Van Horn et al., 2018), Textures (Cimpoi et al., 2014), OpenImage-O (Wang et al., 2022)

**Rank Calculation:** Since the performance of the methods differs between near-OOD, far-OOD and the different benchmarks, we use the benchmarks rank for the evaluation. The estimate a rank, we add the rank of the method for each benchmark and estimate a overall ranking based on the combined rank values

**Full Cycle OOD Experiments:** For the full cycle OOD experiments, we used three seeds as common of OpenOOD of a CIFAR-10 AL cycle. The three seeds are taken from a SISOMe training and used as checkpoint for all OOD detection methods. Besides providing different checkpoints, we followed the OpenOOD benchmark procedure for CIFAR-10 with the same data splits. As the checkpoint corresponded to SISOMe and has seen less data, we only report our SISOMe .

## B.3 Ablation Study

In this section, we highlight the relevant parameters for the ablation study experiments on SISOM . Namely, we examine the Optimal Sigmoid Steepness (OS) and the Reduced Subset Selection (RS) shown in Table 3.

---

[2]`https://zjysteven.github.io/OpenOOD`

Table 10: Parameters for the Ablation Study, Coverage Radius for RS and Search Space of Optimal Sigmoid Steepness. Best performing values in bold.

| Dataset | Coverage Radius | Layer | Sigmoid Search Values |
|---|---|---|---|
| CIFAR-10 | 0.75 | AdaptiveAvgPool2d-1 | **100**, 1000 |
| | | Sequential-3 | **1**, 10, 1000 |
| | | Sequential-2 | **0.001**, 0.1, 1 |
| | | Sequential-1 | **0.001**, 0.1, 1 |
| CIFAR-100 | 5.0 | AdaptiveAvgPool2d-1 | **1**, 50, 100 |
| | | Sequential-3 | **0.1**, 10, 100 |
| | | Sequential-2 | **0.1**, 1 |
| | | Sequential-1 | 0.005, **0.1** |
| ImageNet | 10.0 | AdaptiveAvgPool2d-1 | 10, 100, **3000** |
| | | Sequential-3 | **1**, 10, 300 |
| | | Sequential-2 | **0.1**, 1 |
| | | Sequential-1 | **0.1**, 1 |

In the experiments conducted with RS, a representative subset size of 10% relative to the original training set was used across all experiments. Additionally, the specific distance radius used for the class-wise coverage on CIFAR-10, CIFAR-100, and ImageNet is provided in Table 10. A non-class-wise distance radius was also used in a related approach (Yehuda et al., 2022). For SISOM + RS without OS, the suggested sigmoid steepness parameters (Liu et al., 2024) emphasized in Appendix B.2 were used. For the OS modification, the search space for the optimal sigmoid parameters is presented in Table 10. The parameters fulfilling the minimization of $r_{avg}$ are highlighted in bold.

### B.4    Latent Space Visualization

For the UMAP visualization, we used a SISOM AL checkpoint from the last step for CIFAR-10. For the InD data, we visualized the label and remaining unlabeled samples. For OOD, we followed the categorization of OpenOOD and used Tiny ImageNet (Le & Yang, 2015) as the near-OOD dataset and SVHN (Netzer et al., 2011) as a far-OOD dataset.

