# OpenReview forum: "A Unified Approach Towards Active Learning and Out-of-Distribution Detection"
_TMLR — Accepted by TMLR_

### Review · Reviewer_ey25 · 2025-04-25

**Summary Of Contributions:**

The authors' work explores the connection between Active Learning (AL) and Out of Distribution (OOD) detection, introducing a unified approach for both tasks that leverages their mutual strengths.

**Audience:**

Yes

**Claims And Evidence:**

Yes

**Requested Changes:**

Recently, in https://arxiv.org/abs/2308.14815, accepted at HSCC 2025 (https://hscc.acm.org/2025/accepted-papers/), the authors discuss the first ever application of techniques from the imprecise probability literature to AL, precisely to solve problems similar to the ones that the authors are interested in (AL + OOD). It would be nice to see a small discussion on how the two papers relate.

More in general, we suggest the authors to dig --- for their future endeavors --- in the Imprecise Probabilistic Machine Learning field (see contributions by Hüllermeier, Sale, Caprio, Destercke, Cuzzolin, De Campos, Zaffalon, Denoeux, Williamson, Muandet, Benavoli, Cella, Gong, and Cozman), whose tools are extremely helpful to approach and attack problems similar to the ones presented in this paper.

**Strengths And Weaknesses:**

The paper is clearly written, the images are beautiful, and the experiments seem to prove the effectiveness of SISOM. I must admit that I am not an expert on Active Learning, so I cannot comment on the technical aspects of the work. What I can say is that it definitely is of interest for the TMLR readers, and that the problem studied by the authors is very interesting. As such, I recommend the paper is accepted after minor revision.

The weakness that I was able to detect (not really a weakness of course, but rather a missed related work), is highlighted in the following section.

---

> ### Author Response · Authors · 2025-05-22
> **Rebuttal of Reviewer ey25**
>
> Dear reviewer ey25,
>
> thank you for your valuable feedback and for acknowledging our paper as **clearly written** and our experiments as adequate **to prove the effectiveness of SISOM**. We are delighted that you consider our work **of interest for the TMLR readers** and your recommendation **to accept it with minor revisions**.
>
> ## Requested Change
>
> Thank you for hinting at this work. It is quite interesting how other domains target the same motivation but focus on active learning for reinforcement-learned controllers. We added the suggested work to our related work section.
> We think that imprecise probability could indeed be a valuable inspiration for active learning and OOD detection in perception tasks.
>
> However, it should be mentioned that both have a different focus, and paradigms between AL for perception and reinforcement learning can differ quite a lot. SISOM aims to provide a unified framework for simultaneously performing AL and OOD detection, addressing the core problems of efficient data labeling and robust model operation in dynamic environments. Distinctly, "Distributionally Robust Statistical Verification with Imprecise Neural Networks" (DRSV) focuses on providing statistical verification and performance guarantees for high-dimensional autonomous systems that encounter distribution shifts during deployment with a focus on AL for reinforcement learning.

---

### Review · Reviewer_F3xM · 2025-05-07

**Summary Of Contributions:**

The authors present **Simultaneous Informative Sampling and Outlier Mining (SISOM)** as an approach to unify *active learning* (AL) and *out-of-distribution detection* (OOD) objectives. Its core idea is to adopt existing techniques, enhancing the latent feature space's expressiveness, as the basis for computing a inner-to-outer class distance ratio. In an AL setting, this novel ratio guides the selection of instances between clusters within the latent feature space for labeling. At the same time, it enables the separation of *in-distribution* (ID) and OOD data. Due to the two objectives, the authors perform a twofold evaluation study. On the one hand, they evaluate the AL component of SISOM compared to other (well-known or state-of-the-art) AL strategies across image datasets for classification. On the other hand, they assess the OOD detection component of SISOM by following an established OOD detection benchmark.

**Audience:**

Yes

**Claims And Evidence:**

No

**Requested Changes:**

I would appreciate it if the authors could address the abovementioned *weaknesses* and clarify the following *minor remarks* and *questions*.

**Minor Remarks:**
- The mathematical notation is, from time to time, challenging to follow. For example, in Eq. (5) and (6), the inner-class and outer-class distances are introduced as plain variables. At the same time, they are used as functions in Eq. (10). Another example would be Eq. (3) and (4), where the meaning of $\mathbf{z}_i$ (likely $h_i(\mathbf{x})$) and $\odot$ (likely Hadamard product) is not fully disclosed.
- A few paragraphs would benefit from more references. For example,  the first paragraph in Section 1 makes factual statements about mobile robotic perception without citing works for confirmation.
- There are minor typos, e.g., "Sigmoid Stepness" in Fig. 3.

**Questions:**
- Is there a reason for reporting only OOD results corresponding to one AL checkpoint in Table 2 instead of curves indicating the development of OOD detection performance over the number of AL cycles?
- Does the relative accuracy in the learning curve plots refer to relative improvements over random sampling? For example, if one AL strategy achieves 10.1% accuracy and random sampling only 10% accuracy, does this correspond to a relative improvement of 1%?
- Do you have a potential explanation for why your results in Fig. 7 indicate rather bad performance for TypiClust, while other works [1] have confirmed high performance for CIFAR-10? Is it because the amount of labeled data corresponds to a high-budget scenario as shown in [2]?
- Why do you evaluate AL strategies such as BADGE not in the semi-supervised AL setup?

**References:**
- [1] Bae, Wonho, et al. "Generalized Coverage for More Robust Low-Budget Active Learning." European Conference on Computer Vision (2024).
- [2] Bae, Wonho et al. "Uncertainty Herding: One Active Learning Method for All Label Budgets." International Conference on Learning Representations (2025).

**Strengths And Weaknesses:**

Below, I list the *strengths* and *weaknesses* according to their importance, starting with the most important one and ending with the least important one. I will reconsider my perspective on these issues if I have misunderstood or overlooked any aspect. Looking forward to a constructive discussion!

**Strengths:**
- SISOM addresses a critical yet underexplored problem of label-efficiently training an accurate classification model, simultaneously enabling robust OOD detection.
- The proposed distance ratio has an intuitive motivation and is easy to compute. Moreover, the average ratio of the labeled instances allows for defining a criterion for estimating the feature space's expressiveness, which balances the model uncertainty with the inner-to-outer class ratio. The idea is to trust the model's estimated uncertainties more if the feature space's expressiveness is low.
- SISOM is a modular system extracting expressive instance representations by combining established research concepts, such as concatenating multiple layers' latent spaces and gradient-based weighting of individual layers, as building blocks. The authors demonstrate the benefit of the particular building blocks in an ablation study.
- The experimental details and the provided code allow other researchers to reproduce the experimental results.

**Weaknesses:**
- While I acknowledge the effort of performing an evaluation study covering AL and OOD tasks, there is not "extensive" evidence to reliably assess the performance of SISOM in AL tasks. For example, the current AL setup covers only CIFAR-10, CIFAR-100, and SVHN as three relatively simple benchmark datasets. Moreover, there is no information on how the acquisition size per cycle has been determined. Accordingly, the paper would benefit from following more of the recommendations of established evaluation protocols [1, 2].
- As noted by the authors, the computational complexity can be a bottleneck in AL. Subset selection is a standard procedure to address this issue, and the authors have also studied its impact on SISOM. However, there is no further comparison regarding the selection times of SISOM to those of its competitors. For example, SISOM requires the evaluation of gradients, distance computations for latent feature spaces covering multiple neural network layers, and optimizing the $\alpha$ to maximize the feature space separability.
- Certain design choices are not fully explained. For example, the instance selection is performed via the top-$k$ selection scheme in Eq. (8). However, there is no guarantee that this batch contains diverse instances because many similar instances could lie between clusters in the latent space. So it is unclear is whether there could be issues with redundant instances within a selected batch.

**References:**
- [1] Lüth, Carsten, et al. "Navigating the pitfalls of active learning evaluation: A systematic framework for meaningful performance assessment." Advances in Neural Information Processing Systems (2023).
- [2] Werner, Thorben, et al. "A Cross-Domain Benchmark for Active Learning." Advances in Neural Information Processing Systems (2024).

---

> ### Author Response · Authors · 2025-05-22
> **Rebuttal of Reviewer F3xM - Weaknesses**
>
> Dear Reviewer F3xM,
> thank you for your valuable feedback and for valuing our paper for **addressing a critical yet underexplored problem**, its **intuitive motivation,** and its **modular system**. We hope to have addressed all the weaknesses below.
>
> ### Weaknesses:
>
> **Active Learning evaluation** :
> This is a very important point to mention. We split our answer according to the three mentioned main points, "Justification of selection size",  "Evaluation Protocol" and "Extensiveness of Evidence".
>
> - Selection Size:
> Thank you for pointing out this unclarity. We used selection sizes that have been used by highly cited existing active learning works, (e.g., CIFAR-10 Yoo & Kweon, 2019, Kim et al., 2021, Ash et al., 2020; CIFAR-100 Caramalau et. al. 2021,Kim et al., 2021) which has been hinted at in the main paper but insufficiently mentioned in the Appendix. With our method, we aimed to the mostly benchmarked high data region in which the settings of Yoo & Kweon 2019 and  Kim et al. 2021 align with [1] suggestions for CIFAR-10. For CIFAR-100, Caramalau et. al. 2021 choose 2000, for which [1] suggests 1000 for mid and 5000 for the high data region.
>
> - Evaluation Protocol:
> By following the established setup of highly cited works for our experiments (Only for SVHN we differ from the suggestion of Caramalau et. al. 2021,Kim et al., 2021) we aim for a broad overview of our method.
> While [1] highlights different aspects of active learning scenarios that can be evaluated, most works focus on one data regime (Hacohen et al., 2022) or training paradigm and rely on established parameters. We agree that the suggestions made by [1] can strengthen our active learning evaluation. We additionally evaluated the mid-data regime and conducted a study on selection size as recommended in [1] P2&P3. In addition, we added an other model to our evaluation. As we already covered multiple training paradigms, we now cover a wide variety of the active learning scenario landscape.
>
> - "Extensive" Evidence:
> We admit that our existing evaluation only covered the primary used high-data regime. By extending our experiments to include more of the recommended facets [1] of active learning, we provided additional evidence for our approach for the primarily used datasets. It should be noted that, as shown in [1,2,A], most active learning works have a specific focus that they target and perform well, such as low-data regimes. Given this, providing a solution for every facet is hardly possible.
> Thus, our goal is not to overshadow existing approaches in a variety of huge or complex tasks and datasets with less established experiment settings. Moreover, we focus on showing that SISOM, a joint method design, leads to competitive results in both tasks on well-established benchmarks and well-established and primarily used active learning setups, as well as highlight the ambiguity between (near) OOD and interesting active learning samples.
>
>
> [A] A Comparative Survey of Deep Active Learning, Xueying Zhan, Qingzhong Wang, Kuan-hao Huang, Haoyi Xiong, Dejing Dou, Antoni B. Chan, Arvix 2022
>
> **Computational Complexity**:
> Thank you for pointing this out. Indeed, the runtime is an important attribute. We included an empirical runtime analysis to show how our subset selection improves the temporal behavior of SISOM in Section 4. In addition, we underlined the temporal effectiveness of SISOM compared to other active learning methods and include a comparison against other OOD detection methods in Sec. A. Additionally, we point out that we get parts of the required values for computation of $\hat{z}$ for "free", such as the feature vector $z$, that has to be computed during the forward pass.
>
> **Design Choices**:
> Thank you for pointing out this unclarity. We chose the top-k approach as is the most used approach in active learning, e.g. (Kim et al. 2021, Yoo & Kweon, 2019, Caramalau et al. 2021, Ning et al. 2022) which we would consider the standard approach. We added references to underline this point. Yet it is true, that many active learning approaches categorized as diversity-based have included a mechanism for batch diversity. In fact, the problem of batch diversity is a common problem in most active learning methods and has been investigated by works like (Kirsch et al., 2019). Since we calculate a single score instead of a scoring matrix like Ash. et al. (2020) or Sener and Savarese (2018), it is not straightforward to put a mechanism in place. However, this design choice enables SISOM to operate in the stream-based scenarios (Schmidt and Günnemann 2023) since each sample is evaluated individually.
> By adding a study on different query sizes, we investigate the effects, which is common practice by previous works e.g. Ash. et. al. (2020).

---

> > ### Author Response · Authors · 2025-05-22
> > **Rebuttal of Reviewer F3xM - Minor Remarks and Questions**
> >
> > ### Minor Remarks:
> > **The mathematical notation**
> > Thank you for your hints. We went over all the mathematical formulas to make sure they were aligned. In Eq. (5) and (6), $d_{in}$ and $d_{out}$ should have been defined as functions of a feature space vector. In Eq. (3) and (4), you are correct in your assumptions, and we have added the explanations in the paper.
> >
> > **Few References**:
> > Thank you for your feedback regarding our references, we updated our manuscript with more references.
> >
> > **Typos**
> > Thanks again for pointing us towards this typo, which we took as an occasion to proofread the paper once more.
> >
> > ### Questions:
> >
> > **Q1**:
> > Thank you for this good question and idea. We assumed that the robot would operate after a successful training since we did not cover continuous learning in our work. Especially the early phase of active learning and OOD detection would not be meaningful as no reasonable task performance has been reached.
> >
> > **Q2**:
> > It refers to the absolute performance difference between the method and random selection **not** to a relative one.
> >
> > **Q3**:
> > TypiClust has been mostly evaluated for low data regions for different task regions, where it can leverage the full strength of the pretrained model embeddings. Especially, the authors reported an outperformance in low data region supervised setting where TypiClust has access to pretrained embeddings. For mid to high data regions the authors reported in their followup work ProbCover a drop in performance similar to [2] for mid and high data regimes, which we evaluated.
> > In addition, we provided the pretrained embeddings model as starting checkpoint for all methods to enable a fair compairson.
> >
> > **Q4**:
> > We focus on the semi-supervised evaluation of methods that already use embeddings or information from pre-trained models. TypiClust and Probcover use features of a pre-trained model for selection, while PT4AL performs a pre-task pretraining. However, we agree and add Badge to this experiment to give a broader overview.

---

> > > ### Comment · Reviewer_F3xM · 2025-06-07
> > >
> > > Dear authors,
> > >
> > > many thanks for your detailed rebuttal and the corresponding changes to improve the manuscript, which I will consider as part of my official recommendation. Independent of whether your paper is accepted or not, I would briefly point out minor adjustments to ensure a clear and coherent presentation of your approach:
> > >
> > > **Content:**
> > > - *Extensive Experiments*: I agree that providing a solution for every AL setting and evaluating such a solution is hardly possible. Yet, I would propose to point out your focus and reasoning behind this foucs earlier. For example, in the abstract you could briefly state that your AL evaluation is restricted to image benchmarks by writing
> > > > "In AL, SISOM outperforms ... in common image benchmarks."
> > >
> > >   Indicating such "limitations" early helps to minimize the gap between claims (from a reader's perspective) and evidence.
> > > - *$Q_1$:* I agree that users want to apply their actively trained models once sufficient performance is reached. However, the definition of this sufficient performance and its evaluation in an active learning setup with different application-dependent budgets (cf. challenge of stopping criteria [1]) are themselves a major challenge. Therefore, many works [2, 3] resort to learning curves (as you did for accuracy) or the area under them to evaluate the strategies' performances. Accordingly, I would have expected similar learning curves for the OOD detection performance (maybe with a lower resolution in terms of the active learning cycles to keep the computation complexity limited).
> > >
> > > **Presentation:**
> > > - Fig. 3 does not match the notation in the paper. For example, vectors are not written in bold.
> > > - The score $r$ in Eq. 7 is not defined as a function of an instance, although $d_{\\mathrm{in}}$ and $d_{\\mathrm{in}}$ are now defined as functions.
> > > - The variables $r_i$ and $E_i$ are not properly defined. I expect them to refer to individual instances $\mathbf{x}$.
> > > - Fig. 7 should indicate that the results refer to the relative accuracy gains compared to random sampling as percentage points (similar to Fig. 6 and 7). In both case, the y-label could also refer to $[\\%\\,\\mathrm{pt.}]$ or [p.p.] as unit.
> > > - Double-check the spelling and consistent writing of the entire text, e.g., "N~n~ear-OOD" in the caption of Table 1 and "datasets.Given" in Table 5 in the Appendix,
> > > - Table 4 does not provide the time unit. I expect it to be recorded in seconds since Appendix A.6 mentions it.
> > > - Resolving remaining issues with the references, e.g., the paper "On Out-of-distribution Detection with Energy-based Models" is referenced as arXiv paper, although it was published at the "ICML 2021 Workshop on Uncertainty and Robustness in Deep Learning".
> > >
> > > **References:**
> > > - [1] Ishibashi, Hideaki, and Hideitsu Hino. "Stopping criterion for active learning based on deterministic generalization bounds." In AISTATS, 2020.
> > > - [2] Zhan, Xueying, Huan Liu, Qing Li, and Antoni B. Chan. "A Comparative Survey: Benchmarking for Pool-based Active Learning." In IJCAI, pp. 4679-4686. 2021.
> > > - [3] Zhan, Xueying, Qingzhong Wang, Kuan-hao Huang, Haoyi Xiong, Dejing Dou, and Antoni B. Chan. "A comparative survey of deep active learning." arXiv preprint arXiv:2203.13450 (2022).

---

> > > > ### Author Response · Authors · 2025-06-12
> > > > **Response to Rebuttal Response and Minor Points**
> > > >
> > > > Dear reviewer F3xM,
> > > >
> > > > Thank you for your valuable response and suggestions.
> > > >
> > > > We updated our manuscript with a more precise description of the AL benchmark in the abstract and introduction.
> > > >
> > > > Regarding Q1, we understand your perspective. However, since evaluating OOD detection after AL is a novel task, there are no best practices to follow precisely. Reporting accuracy is a common practice in AL and directly aligns with the goal of the selection strategy, reflecting the state of a continuous cycle. In contrast, OOD detection is not the primary goal of the selection strategy. In our setup design, the deployment would be an interruption, and ODD detection is usually reported in tables, which we did. However, considering the issue of stopping criteria, we added a plot for the last half of the cycles, which could be reasonable candidates, as shown in the appendix.
> > > >
> > > > In addition, we went through your presentation topics and addressed them. Please note that the mentioned literature example "On Out-of-distribution Detection with Energy-based Models" was presented at the ICML 2021 Workshop on Uncertainty and Robustness in Deep Learning. However, the workshop does not publish proceedings, thus we cited arvix. We can change the citation style for workshops without proceedings for a camera-ready version.
> > > >
> > > > Since you provided your feedback "Independent of whether your paper is accepted or not" and you are the only reviewer who did not mention their recommendation for acceptance, we politely would like to know if we can clarify further points, hopefully leading to a consensus on acceptance.
> > > >
> > > > Sincerely,
> > > >
> > > > TMLR Paper4629 Authors

---

### Review · Reviewer_vovk · 2025-05-08

**Summary Of Contributions:**

The paper proposes a method for active learning and OOD detection. The method calculates the distance ratio of the test sample and the closest labeled samples coming from the same/different class as the predicted class of the test sample. Active learning selection is then performed by selecting samples with the highest distance ratio. On the other hand, the OOD detection is performed by mapping the distance ratio to the $[0,1]$ range and then thresholding. The method is evaluated on three benchmarks for active learning and three benchmarks for OOD detection, and the proposed method performs well across all of them.

**Audience:**

Yes

**Broader Impact Concerns:**

The work does not have any ethical concerns that would need to be addressed in a broader impact statement.

**Claims And Evidence:**

Yes

**Requested Changes:**

*Critical*
- The paper should include the comparisons with the missing state-of-the-art OOD detection methods and fix the discrepancies with the official results from the OpenOOD benchmark, as mentioned in the weaknesses.
- The paper should fix the claims about all OpenSet Active Learning methods using separate modules for active learning and OOD detection, and include the discussion of existing works that use a single module.
- The paper should include the running time of the proposed method and compare it with the existing methods for active learning and OOD detection.

*To strengthen the work*
- The paper should evaluate the proposed active learning method on larger benchmarks, e.g., ImageNet, and using larger backbones, e.g., ResNet50.
- The paper should include the absolute performance in the active learning experiments shown in Figure 6 and Figure 7, as showing only the difference to random selection does not show whether the trained models are performing well.

**Strengths And Weaknesses:**

**Strengths**

- The method brings together two disjoint fields: active learning and OOD detection. These two fields share a similar goal of detecting data different from the data seen during training, but doing it for different reasons. Active learning postulates that data different from the seen one is more useful for future model training. On the other hand, OOD detection wants to filter out the data outside of the model's distribution.
- The method presented in the paper is the first one that operates in both tasks and achieves competitive results.
- The proposed method is simple, and the design choices are well motivated.
- The paper is clearly written and easy to follow.

**Weaknesses**
- Evaluation of active learning experiments could be improved.
	- Active learning is evaluated only on tiny datasets such as CIFAR and SVHN, while recently, many active learning methods [1, 2, 3] are moving their evaluation to more realistic datasets such as ImageNet. It would be beneficial to show that the proposed method performs well in this realistic setup.
	- The method is evaluated only using the ResNet18 backbone. Although this is the most common backbone in active learning, it would be good to show the performance of the proposed method with larger backbones, e.g. ResNet50, as used in recent active learning methods [1].
- The paper is missing a comparison with some state-of-the-art OOD detection methods.
	- Some of these are SCALE [4], CombOOD [5], and NNGuide [6].
	- Additionally, the performance of some of the competitors in near-OOD is different than what is reported on the official OpenOOD benchmark website, e.g. NAC for ImageNet (71.73 vs 74.43).
- The paper claims that OpenSet Active Learning methods tackle active learning selection and OOD detection with separate modules, which is not true for all methods. Two recent methods [3, 7] handle the OOD detection and classification using the same model. OOD detection simply uses an additional class in the classifier that handles all other classes.

*References*

[1] Guy Hacohen, Avihu Dekel, Daphna Weinshall, "Active Learning on a Budget: Opposite Strategies Suit High and Low Budgets", In ICML, 2022

[2] Dongmin Park, Yooju Shin, Jihwan Bang, Youngjun Lee, Hwanjun Song, Jae-Gil Lee, "Meta-Query-Net: Resolving Purity-Informativeness Dilemma in Open-set Active Learning", In NeurIPS, 2022

[3] Vladan Stojnić, Zakaria Laskar, Giorgos Tolias, "Training Ensembles with Inliers and Outliers for Semi-supervised Active Learning", In WACV, 2024

[4] Kai Xu, Rongyu Chen, Gianni Franchi, Angela Yao, "Scaling for Training Time and Post-hoc Out-of-distribution Detection Enhancement", In ICLR, 2024

[5] Magesh Rajasekaran, Md Saiful Islam Sajol, Frej Berglind, Supratik Mukhopadhyay, Kamalika Das, "COMBOOD: A Semiparametric Approach for Detecting Out-of-distribution Data for Image Classification", In Proceedings of the International Conference on Data Mining, 2024

[6] Jaewoo Park, Yoon Gyo Jung, Andrew Beng Jin Teoh, "Nearest Neighbor Guidance for Out-of-Distribution Detection", In ICCV, 2023

[7] Suraj Kothawade, Nathan Beck, Krishnateja Killamsetty, Rishabh Iyer, "SIMILAR: Submodular Information Measures Based Active Learning In Realistic Scenarios", In NeurIPS, 2021

---

> ### Author Response · Authors · 2025-05-22
> **Rebuttal of Reviewer vovk**
>
> Dear Reviewer vovk,
>
> thank you for your insightful feedback on our paper and for recognizing our efforts to **bringing AL and OOD detection together** in a **simple and well motivated** approach, which is **the first to achieve competitive results in both tasks**. In addition, we are grateful that our paper is perceived as **clearly written** and **easy to follow**. We adapted all critical requests as described below and strengthened our paper as suggested.
>
>
> ## Weaknesses and requested changes:
> ### Critical
>
> - **OOD Scores and Baselines**:
> We apologize for having a wrong score reported in our paper; we seem to have used a code version based on the first preprint draft of the respective paper. We corrected the score and double-checked all other sources.
> We added the three requested baselines to our paper. For SCALE and CombOOD, we report the scores identical to the online OpenOOD benchmark leaderboard. For NNGuide, which was not reported for the investigated settings, we added the results to Tables 1 and 5 in the Appendix. Given the large number of **18** baselines with different focus and strengths, we estimated an aggregated global rank based on the individual baseline ranking of the methods in which SISOMe ranks **first** and is the only method with top three rankings on **all benchmarks**.
>
> - **OSAL Claims**: We thank the reviewer for pointing out these works. In fact, we seem to have generalized the properties without them. It is correct that both works do not have two distinct components like, for example, LfOSA (Ning et al. 2022), but it should be mentioned that [3] follows the idea of LfOSA of an open set class but integrates them into an ensemble of multiple models and a semi-supervised learning framework with two phases. SIMILAR, in contrast, builds upon submodular optimization and requires access to initial known OOD data to perform a computationally costly optimization ([3] reported memory problems for SIMILAR).
> We added both related works and included an additional discussion, which also included the number of used models, the used data access, and the training mode, which has been simplified in our work.
>
> - **Runtime**:
> Thank you for this valuable suggestion. We added a runtime table for different subset sizes in the main paper and additional tables for a comparison with the OOD and AL learning methods in the Appendix. It can be seen that our subset selection massively reduces the run time of SISOM.
>
> ### To strengthen the paper:
> - **Additional Benchmarks on models or datasets**:
> With our focus on well-established experiment settings (which include initial set, query sizes, and model selection, e.g., CIFAR-10 -Yoo et al., Kim et al., 2021 Caramalau et al., 2021 ), which are present in most existing works to show the broad applicability of unifying OOD detection and active learning. With our additional experiment on other data regimes, query sizes, and **larger models**, we underlined the versatility of SISOM for different active learning domains and data regimes. While other papers started using larger datasets like ImageNet, they mostly use custom variations of class subsets [1,2,3], resize the set [2] or/and evaluate the (very) low data regime [1], as the dataset itself is too computational demanding for active learning and tailored to the address gap of the paper, which would only limitedly underline the versatility of our concept since we more focus on a broad overview than a specific gap.
>
> - **Reporting Absolute numbers**:
> Reporting the relative number is, in fact, a common practice in active learning papers (e.g., Kim et. al.2021). Since we built on established active learning experiment parameter choices (Caramalau et al. 2021, Yoo & Kweon, 2019), we did not consider this point and understand your concerns.
> We reported some of the additional experiments added as described above with absolute numbers, given that these scenarios are also more seldomly used. If requested, we can also add an absolute number version for other experiments to the Appendix.

---

> > ### Comment · Reviewer_vovk · 2025-06-04
> > **Rebuttal answer**
> >
> > Dear Authors,
> >
> > Your answer has addressed all of my major concerns from the initial review. I agree that performing active learning experiments on ImageNet is very expensive and possibly not feasible. I am glad you could at least perform the experiments with larger backbones (ResNet-34), which indicates that the proposed method is scalable. As for the absolute numbers, I do not think it is necessary to report them for all active learning experiments. However, it is good that you have added them to some experiments as they indicate the overall performance of different methods. Without them, a reader would be unable to see if improvements are only happening because performance is very low.
> >
> > Additionally, I have read other reviews and replies. The authors have addressed all major comments that were raised adequately.
> >
> > Based on all of this, I am willing to recommend the acceptance.
> >
> > Reviewer vovk

---

> > > ### Author Response · Authors · 2025-06-12
> > > **Thank you for your response**
> > >
> > > Dear reviewer vovk,
> > >
> > > thank you for your response. We are pleased to have answered all your questions and appreciate your consideration of our paper for acceptance.
> > >
> > > Sincerely,
> > >
> > > TMLR Paper4629 Authors

---

### Author Response · Authors · 2025-05-22
**General Rebuttal Infos**

We greatly appreciate the effort of the reviews and their valuable feedback.
We are particularly encouraged by your recognition of several strengths in our work, such as providing an **integration of Active Learning and OOD Detection**, the novelty and simplicity of our method as **the first to operate across both tasks with competitive results**, and addressing a **critical yet underexplored problem**. We are delighted that our **intuitive motivation**  and **modular system** design was appreciated.
Lastly, we are glad that our manuscript convinced with its **clearance of presentation** and its **nice figures**.

We updated our manuscript to include your change request and marked all changes in blue (except completely new or reworked tables or figures have only blue captions).
We address all your concerns below and hope for a fruitful discussion.

---

### Decision · Action_Editor_reFF · 2025-07-17

**Recommendation:** Accept as is

**Additional Comments:**

The reviewers ask that the authors include the appendix at the end of the main paper for better visibility. Also, we recommend another pass over the papers to clean up some minor points in the camera ready, e.g.

page 4 typo: 'overlap n distribution'
page 11 typo: 'SISOMe archived' -> (achieved?)
Table 1: how are the rankings aggregated?
Table 2: for other methods such as ReAct, does it also use the AL check points of SISOM?
Table 1: is the AUROC of 81.10 for SISOMe on CIFAR 100 meant to be equal to that of Table 4 with 5% subsampling?
Appendix B.2, sigmoid values means sigmoid steepness parameters?

**Audience:**

Yes

**Audience Explanation:**

Both active learning and OOD detection are research problems that are of good interest to the TMLR community.

**Claims And Evidence:**

Yes

**Claims Explanation:**

The reviewers appreciated the finding that a novel notion of distance ratio between in-class and out-of-class can serve as uncertainty metrics for both active learning and OOD detection. They commended about the simplicity of the proposed approach.

For each problem (active learning and OOD detection), the paper conducted experimental comparison between the proposed algorithms and 10+ baselines in image classification datasets and OpenOOD datasets.